# A Data-driven Typology of Vision Models from Integrated Representational Metrics

## Abstract

Large vision models differ widely in architecture and training paradigm, yet we lack principled methods to determine which aspects of their representations are shared across families and which reflect distinctive computational strategies. We leverage a suite of representational similarity metrics, each capturing a different facet—geometry, unit tuning, or linear decodability—and assess family separability using multiple complementary measures. Metrics preserving geometry or tuning (e.g., RSA, Soft Matching) yield strong family discrimination, whereas flexible mappings such as Linear Predictivity show weaker separation. These findings indicate that geometry and tuning carry family-specific signatures, while linearly decodable information is more broadly shared. To integrate these complementary facets, we adapt Similarity Network Fusion (SNF), a method inspired by multi-omics integration. SNF achieves substantially sharper family separation than any individual metric and produces robust composite signatures. Clustering of the fused similarity matrix recovers both expected and surprising patterns: supervised ResNets and ViTs form distinct clusters, yet all self-supervised models group together across architectural boundaries. Hybrid architectures (ConvNeXt, Swin) cluster with masked autoencoders, suggesting convergence between architectural modernization and reconstruction-based training. This biology-inspired framework provides a principled typology of vision models, showing that emergent computational strategies—shaped jointly by architecture and training objective—define representational structure beyond surface design categories.

## 1 Introduction

The rapid proliferation of vision models—spanning diverse architectures from CNNs to Vision Transformers, and training paradigms from supervised to self-supervised learning—has created a rich ecosystem of computational approaches to visual processing. Yet we lack principled methods to understand which aspects of learned representations are universally shared across this diverse landscape and which are specific signatures of particular model families. Do all vision models converge on similar geometric organizations of their representation spaces? Is linearly accessible information a common currency across architectures, or do different model families encode information in fundamentally distinct ways? These questions are critical for understanding the computational principles underlying successful vision models and for predicting how different models will behave on novel tasks.

Current approaches to model comparison rely heavily on individual similarity metrics—Representational Similarity Analysis (RSA) (Kriegeskorte et al.), Centered Kernel Alignment (CKA) (Kornblith et al., 2019), Linear Predictivity (Yamins et al., 2014), and others—each capturing a specific facet of representational structure. However, this fragmented methodology obscures a crucial insight: different representational facets (geometry, tuning properties, linearly accessible information) may vary in their universality across model families. Some aspects might reflect convergent computational solutions shared broadly across architectures, while others might constitute distinctive signatures of specific model families. Understanding this landscape requires not just comparing models, but systematically evaluating which representational properties are conserved versus specialized. Furthermore, relying on a single representational similarity metric is inherently fragile and can easily enable cherry-picking. A growing body of work shows that commonly used similarity metrics can yield contradictory views of model relatedness: tuning-sensitive

metrics (e.g., SoftMatch) distinguish architectural families, whereas geometry-preserving measures (e.g., CKA/RSA) are more sensitive to training vs. random initialization but often conflate architectures(Bo et al., 2024). Because each metric emphasizes different invariances, no single metric provides a complete characterization, making conclusions highly dependent on metric choice. This motivates the need for a principled fusion method that integrates complementary representational signals to obtain a more stable and robust organization of model space. In this work, we introduce a framework that addresses these challenges through two key contributions. First, we systematically evaluate how different representational facets discriminate between model families, revealing that geometry or tuning-preserving metrics (RSA, Soft Matching (Khosla & Williams, 2024)) strongly separate families while metrics capturing linearly accessible information show weaker discrimination. This pattern indicates that tuning of individual neurons is family-specific and that linearly accessible signals vary less across these model families. Second, inspired by multi-omics approaches in biology where diverse molecular signatures are integrated to reveal cell types, we employ Similarity Network Fusion (SNF) (Wang et al., 2014) to combine these complementary perspectives into unified model signatures that provide clearer family identity than any single metric alone.

This integrated approach enables us to construct a **data-driven typology of vision models**—a novel contribution that moves beyond surface-level architectural categories to reveal how models actually organize information. Our typology does not rely on a priori assumptions about which models should group together based on architecture or training method. Instead, following empirical traditions in psychology, neuroscience, and genetics (Wang et al., 2014; Letwin et al., 2006; Echtermeyer et al., 2011; Mukamel & Ngai, 2019) where researchers identify clusters of individuals through correlations across multiple behavioral or molecular indices, we discover natural groupings based on how models process visual information. While typologies could alternatively be defined from theoretical perspectives emphasizing explicit model properties like architecture or training data, our empirical approach reveals surprising patterns: all self-supervised models form a unified cluster that transcends architectural boundaries, with self-supervised ResNets grouping more closely with self-supervised ViTs than with their supervised architectural siblings. Similarly, hybrid architectures (ConvNeXt, Swin) (Liu et al., 2022; 2021) cluster with MAE models, suggesting that architectural modernization and masked reconstruction converge on similar computational strategies despite different design origins.

This data-driven typology provides researchers with a reference framework for understanding where any model instance sits within the broader landscape of vision models. Just as biological taxonomies help scientists understand relationships between species based on genetic and phenotypic traits, our representational typology reveals the "species" of vision models based on their computational strategies. By developing methods to systematically integrate different facets of representation in comparative analyses of models, we provide practical tools for navigating the expanding universe of vision models—enabling researchers to understand model relationships, predict transfer learning compatibility, and make informed choices about which models will exhibit similar behaviors on novel tasks.

## 2 METHODS

### 2.1 MODEL SELECTION AND DATASET

We define the model family as the combination of training paradigm and architecture. We analyze 35 vision models across four primary categories: supervised Convolutional Neural Networks (CNNs), self-supervised CNNs, supervised Transformers, and self-supervised Transformers. We treat ConvNeXt (Liu et al., 2022) and Swin (Liu et al., 2021) as distinct families due to their hybrid nature—ConvNeXt incorporates Transformer-inspired design principles within a convolutional architecture, while Swin introduces CNN-like inductive biases into the Transformer framework. We deliberately restricted the model set to encoders pre-trained on ImageNet-1k. This design isolates the effects of architecture and training objective within a shared data and label space, so that differences in training data distribution do not become a dominant confound. For datasets, we use the ImageNet-1k (Deng et al., 2009) and Ecoset (Mehrer et al., 2021) validation sets and the CIFAR10 (Krizhevsky, 2009) and CIFAR100 (Krizhevsky, 2009) test sets. Complete dataset and model details are provided in Appendix A and B.

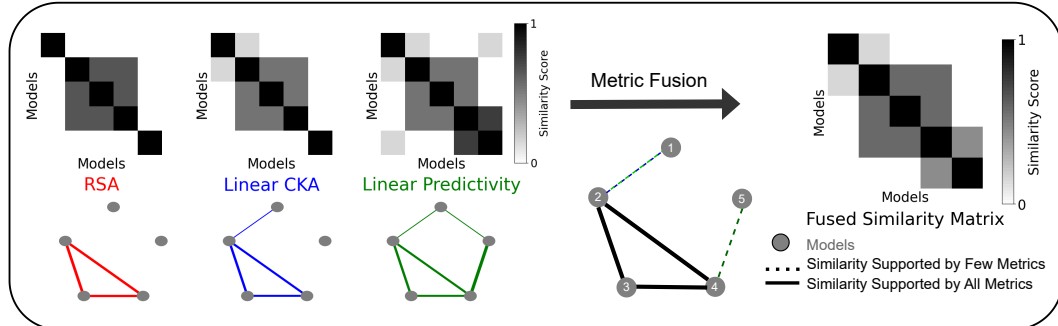

Figure 1: **Top left:** Each representational metric defines a pairwise similarity matrix over models. **Bottom left:** Each matrix is visualized as an affinity graph, with nodes representing models and edge widths reflecting pairwise similarity strength; weak similarities below a threshold are omitted for clarity. **Right:** A consensus matrix obtained via Similarity Network Fusion (SNF) highlights relations consistently supported across metrics while leveraging complementary signals. In the fused graph, solid edges denote agreement across all metrics, dotted edges indicate partial support; strong but uncorroborated edges may persist with reduced weight (e.g., edge 4–5); weak AND metric-specific connections are typically suppressed (e.g., edge 1–5).

## 2.2 REPRESENTATIONAL METRICS

We evaluate widely used similarity metrics that differ in the flexibility of the mappings they permit—from permutation-based alignments (soft-matching) to rigid geometric transformations (Procrustes) to looser linear mappings (linear predictivity) as well as non-fitting approaches that compare representational geometry directly (RSA). Consider two representations $\mathbf{X}_i \in \mathbb{R}^{M \times N_i}$ and $\mathbf{X}_j \in \mathbb{R}^{M \times N_j}$ from different models, where $M$ denotes the number of stimuli and $N_i, N_j$ denote the number of units. All representations are mean-centered along the sample dimension as required. For metrics requiring a fitting procedure (e.g., Soft matching, linear predictivity, Procrustes), similarity values reflect the mean 5-fold cross-validation score.

**Singular Vector Canonical Correlation Analysis (Raghu et al., 2017).** SVCCA first applies singular value decomposition (SVD) to the representation matrices from two models or layers to isolate their most informative directions: $\mathbf{X}_i = \mathbf{U}_i \Sigma_i \mathbf{V}_i^\top$ and $\mathbf{X}_j = \mathbf{U}_j \Sigma_j \mathbf{V}_j^\top$. Retaining the top $N_i'$ and $N_j'$ singular vectors that explain 99% of the variance yields the reduced representations: $\mathbf{X}_i' = \mathbf{U}_i^{(N_i')\top} \mathbf{X}_i$ and $\mathbf{X}_j' = \mathbf{U}_j^{(N_j')\top} \mathbf{X}_j$ whose dominant singular directions capture a disproportionate share of the total information. Canonical correlation analysis (CCA) (Hardoon et al., 2004) is then applied to these reduced matrices to find linear projections $\mathbf{A}$ and $\mathbf{B}$ that maximize their correlation: $\mathbf{Q} = \max_{\mathbf{A}, \mathbf{B}} \text{corr}\left(\mathbf{A}\mathbf{X}_i', \ \mathbf{B}\mathbf{X}_j'\right)$ subject to unit-variance constraints, providing a final similarity score that reflects how closely the informative subspaces of the two representations align.

**Projection-Weighted Canonical Correlation Analysis (Morcos et al., 2018).** Compared to SVCCA, PWCCA did not apply SVD before CCA and it re-weights the canonical directions according to their contribution to the original representation. After CCA, we obtain canonical vectors $\mathbf{A}$ and $\mathbf{B}$ and their corresponding correlations $q_i$ for $i = 1, 2, ...k$, where $k = \min(N_i, N_j)$. Instead of giving each direction equal weight, PWCCA projects the unreduced representation $\mathbf{X}_i$ onto its own canonical vectors to measure how strongly each one reconstructs the data. The projection weight for the $i^{th}$ canonical direction is $\alpha_i = \frac{\|\mathbf{X}_i \boldsymbol{a}_i\|_1}{\sum_{j=1}^{k} \|\mathbf{X}_i \boldsymbol{a}_j\|_1}$ where $\boldsymbol{a}_i$ is the $i^{th}$ column of $\mathbf{A}$. The final PWCCA similarity is then the weighted sum of canonical correlations: $\Sigma_{i=1}^{k} \alpha_i q_i$ which emphasizes directions that explain the greatest fraction of variance in $X_i$ and down-weights noisy, low-variance components, yielding a more faithful measure of representational similarity than the equal-weight SVCCA score.

**Linear Centered Kernel Alignment (Kornblith et al., 2019).** Linear CKA provides a scalar measure of how similarly two sets of representations capture the relationships among the same

collection of samples. It is is defined as $\frac{\|\mathbf{X}_i^T\mathbf{X}_j\|_F^2}{\|\mathbf{X}_i^T\mathbf{X}_i\|_F\|\mathbf{X}_j^T\mathbf{X}_j\|_F}$, where $\|.\|_F$ is the frobenius norm, and $\mathbf{X}_i$ and $\mathbf{X}_j$ can be assumed to be normalised. Linear CKA is invariant to orthogonal transformations (rotations or reflections), isotropic scaling, and translations of the representations, so it captures only the relational structure shared between the two spaces.

**Representational Similarity Analysis (Kriegeskorte et al.).** Representational Similarity Analysis (RSA) compares the geometry of representations via their Representational Dissimilarity Matrices (RDMs). For each representation, we compute pairwise dissimilarities between stimuli using $1 - $ Pearson correlation, yielding an $M \times M$ RDM that encodes the relational structure. Model similarity is then quantified as the Pearson correlation between their RDMs. RSA is invariant to orthogonal transformations and reflects how models structure their representational spaces, independent of the specific features they encode.

**Soft Matching (Khosla & Williams, 2024).** Soft Matching (SoftMatch) generalizes permutation distance (Ding et al., 2021) to representations with different numbers of units by relaxing permutations to "soft permutations." Specifically, consider a non-negative matrix $\mathbf{P} \in \mathbb{R}^{N_i \times N_j}$ whose rows each sum to $1/N_i$ and columns to $1/N_j$. The set of such matrices defines a transportation polytope (De Loera & Kim, 2013), $\mathcal{T}(N_i, N_j)$. The optimization problem is

$$d_T(\mathbf{X}_i, \mathbf{X}_j) = \min_{\mathbf{P}\in\mathcal{T}(N_i,N_j)} \sum_{k,l} \mathbf{P}_{kl} \|x_i^{(k)} - x_j^{(l)}\|^2,$$

where $x_i^{(k)}$ and $x_j^{(l)}$ are the $k$-th and $l$-th columns (units) of $\mathbf{X}_i$ and $\mathbf{X}_j$. The optimal transport plan $\mathbf{P}^\star$ is found via the network simplex algorithm. When $N_i = N_j$, this reduces to an optimal permutation. The final similarity score is the mean unit-wise correlation between $\mathbf{X}_j$ and $\mathbf{X}_i\mathbf{P}^\star$.

**Procrustes Alignment (Ding et al., 2021).** Procrustes analysis finds the orthogonal transformation that best aligns two representations while preserving geometry. For unequal dimensions, the smaller representation is zero-padded. The optimization problem is $\min_{\mathbf{R}\in\mathcal{O}(N)} \|\mathbf{X}_j - \mathbf{X}_i\mathbf{R}\|_2^2$, where $\mathcal{O}(N) = \{\mathbf{R} \in \mathbb{R}^{N\times N} : \mathbf{R}^\top\mathbf{R} = \mathbf{I}\}$. The optimal transformation $\mathbf{R}^\star$ is obtained via singular value decomposition. The similarity score is the mean unit-wise correlation between $\mathbf{X}_j$ and $\mathbf{X}_i\mathbf{R}^\star$.

**Linear Predictivity (Yamins et al., 2014).** Linear predictivity seeks an unconstrained linear transformation that best predicts one representation from another: $\min_{\mathbf{L}} \|\mathbf{X}_j - \mathbf{X}_i\mathbf{L}\|_2^2$. The optimal mapping $\mathbf{L}^\star$ is estimated via ordinary least squares. The final similarity score is the mean unit-wise correlation between $\mathbf{X}_j$ and $\mathbf{X}_i\mathbf{L}^\star$.

**Average Baseline.** To provide a baseline that naively uses all metrics' information, we symmetrized and min-max rescaled all metrics' result matrices and simply averaged them.

## 2.3 SEPARATION ABILITY METRICS

We next describe the measures used to evaluate how well representational metrics capture separability between model families. Because family separation is inherently bidirectional, we compute directional scores in both directions and report the average as the final result.

**Contrastive Ratio.** The contrastive ratio quantifies the relative separation between intra-family and inter-family similarities. We consider the similarity values between different models within the same family and take the average of them to obtain $\mu_{\text{within}}$, and the similarities between models from two model families and obtain the average inter-family $\mu_{\text{between}}$. The ratio is then defined as $CR = (\mu_{\text{within}} - \mu_{\text{between}})/(\mu_{\text{within}} + \mu_{\text{between}})$. A value approaching 1 suggests strong within-family coherence relative to cross-family similarity; a value approaching 0 suggests no difference, and a negative value implies that inter-family similarity exceeds intra-family similarity.

**D-Prime (Bo et al., 2024).** Similarly to contrastive ratio but considering the variance, the D-Prime ($d'$) also quantifies the separation between intra-family and inter-family similarity distributions. It is defined as $d' = (\mu_{\text{within}} - \mu_{\text{between}})/\sqrt{0.5(\sigma_{\text{within}}^2 + \sigma_{\text{between}}^2)}$, where $\mu$ and $\sigma^2$ denote the mean and variance of the respective distributions. Higher values indicate tighter clustering within a family and greater spread across families, reflecting stronger separability.

**Silhouette Score (Rousseeuw, 1987).** For each model $i$, we compute the average distance $a(i)$ to all other models in the same family and the average distance $b(i)$ to models in the other family. The silhouette value is then $s(i) = (b(i) - a(i))/\max\{a(i), b(i)\}, s(i) \in [-1, 1]$. Values near 1 indicate that the model is well grouped with its own family, values near 0 suggest boundary placement, and negative values imply greater similarity to another family. The overall silhouette score is obtained by averaging $s(i)$ across all models.

## 2.4 SIMILARITY NETWORK FUSION

Next, we sought to reconcile the results across the different evaluation metrics. As elaborated in Section 2.3, each metric captures distinct aspects of model representations and varies in its ability to differentiate between model families. To integrate these metrics, we adopt a unified approach inspired by Similarity Network Fusion (Wang et al., 2014; Markello, 2020). Let $n$ be the number of models, and $\mathbb{V}$ be the set of the representational metrics. For each metric $v \in \mathbb{V}$, we can get a similarity matrix $\mathbf{S}^v \in \mathbb{R}^{n \times n}$, where each entry $\mathbf{S}^v_{ij}$ measures the similarity between the model $i$'s representation and model $j$'s according to metric $v$, as described in Section 2.2. Then, for each metric $v$, we first convert pairwise scores into a dissimilarity matrix $\mathbf{Q}^v$ as this equation, $Q^v_{ij} = 1_{i \neq j}\big(1 - (S^v_{ij} + S^v_{ji})/2\big)$. We then build an affinity $\mathbf{W}^v \in \mathbb{R}^{M \times M}$ with a scaled exponential kernel:

$$\mathbf{W}^v(i, j) = \frac{1}{\sqrt{2\pi(\sigma^v_{ij})^2}} \exp\left(-\frac{(\mathbf{Q}^v_{ij})^2}{2(\sigma^v_{ij})^2}\right), \text{ with } \quad \sigma^v_{ij} = \mu \cdot \frac{\overline{\mathbf{Q}}^v(i,N_i) + \overline{\mathbf{Q}}^v(j,N_j) + \mathbf{Q}^v_{ij}}{3}.$$

Here, $\overline{\mathbf{Q}}^v(i, N_i)$ denotes the average dissimilarity from $i$ to its $K$ nearest neighbors $N_i$ under metric $v$. We set the hyperparameter $\mu \in (0, 1)$ to 0.5 and $K$ to 5 following the original paper.

We view each $\mathbf{W}^v$ as a weighted graph and aim to fuse them into a single matrix that emphasizes relationships consistently supported across metrics while suppressing spurious ones. Following the implementation, we form a row-normalized full matrix and a KNN-sparse matrix for each metric:

$$\mathbf{C}^v_{ii} = \sum_j \mathbf{W}^v_{ij}, \quad \widetilde{\mathbf{W}}^v = (\mathbf{C}^v)^{-1}\mathbf{W}^v, \quad \widehat{\mathbf{W}}^v = \tfrac{1}{2}(\widetilde{\mathbf{W}}^v + (\widetilde{\mathbf{W}}^v)^\top),$$

$$\mathbf{S}^v_{ij} = \widehat{\mathbf{W}}^v_{ij} / \sum_{k \in N_i} \widehat{\mathbf{W}}^v_{ik}, \text{ if } j \in N_i; \quad \text{else } 0.$$

We then run the SNF message-passing updates with diagonal regularization, which keeps self-affinity dominant while improving numerical stability. Initialize $\mathbf{P}^{(v)}_0 = \widehat{\mathbf{W}}^v$. For $t = 0, \ldots, T-1$:

$$\mathbf{P}^{(v)}_{t+1} = \mathcal{B}_\alpha\Big(\mathbf{S}^{(v)}\Big(\tfrac{1}{|\mathbb{V}|-1}\sum_{u \neq v} \mathbf{P}^{(u)}_t\Big)\mathbf{S}^{(v)\top}\Big), \quad \mathcal{B}_\alpha(\mathbf{X}) = \tfrac{1}{2}(\mathbf{X} + \mathbf{X}^\top) + \alpha\mathbf{I}$$

After $T$ iterations, we average the networks and perform a row normalization and symmetrization:

$$\mathbf{P} = \tfrac{1}{|\mathbb{V}|}\sum_{v \in \mathbb{V}} \mathbf{P}^{(v)}_T, \quad \mathbf{D}_{ii} = \sum_j \mathbf{P}_{ij}, \quad \widetilde{\mathbf{P}} = \mathbf{D}^{-1}\mathbf{P}, \quad \widehat{\mathbf{P}} = \tfrac{1}{2}\big(\widetilde{\mathbf{P}} + \widetilde{\mathbf{P}}^\top + \mathbf{I}\big).$$

Lastly, to form a dendrogram for model typology, we cluster the fused affinity $\widetilde{\mathbf{P}}$ with hierarchical clustering using SciPy's linkage function (Virtanen et al., 2020).

## 3 RESULTS

### 3.1 DIVERGENT VERSUS CONVERGENT DIMENSIONS ACROSS MODEL FAMILIES

To understand which aspects of learned representations are universally shared across vision models and which constitute family-specific signatures, we systematically evaluate multiple representational facets. Each metric captures a distinct dimension of representational structure, allowing us to identify where model families converge versus diverge. To assess family discriminability, we quantified the difference between within-family and across-family representational similarities using the separability measures introduced above. The results presented here are based on ImageNet; consistent patterns were also observed on other datasets (Appendix C). Our systematic evaluation reveals that different representational dimensions show markedly different patterns of convergence versus divergence across model families (Figures 2, 3, D.1, D.2, D.3). This variation suggests that while some

computational strategies are universally adopted across architectures and training paradigms, others constitute distinctive signatures of specific model families.

Metrics that preserve representational geometry or unit-level tuning properties demonstrate the strongest ability to discriminate between model families. RSA, which captures representational geometry, achieves the highest separability with a $d'$ of 3.95 and silhouette coefficient of 0.56—indicating that geometric organization or relational structure—how models arrange points in representation space—constitutes a strong family-specific signature. Linear CKA also shows strong discrimination ($d' = 3.91$), which aligns with recent theoretical work showing that centered RSA and linear CKA are mathematically equivalent when appropriate centering is applied (Williams, 2024). Similarly, SoftMatch, which preserves individual unit tuning while mapping two representations, shows robust discrimination ($d' = 3.59$, silhouette = 0.30). Even supervised and self-supervised variants within the same architecture family (particularly CNNs) are reliably separated by these metrics, demonstrating that the training paradigm fundamentally shapes the geometry and tunings of individual neurons, such that metrics diagnostic of these representational facets achieve good separation. These properties thus constitute the unique representational "fingerprints" of model families.

Interestingly, Procrustes alignment—which allows orthogonal transformations—shows intermediate discrimination ($d' = 2.96$), falling between SoftMatch and Linear Predictivity. This reveals a clear pattern among mapping-based metrics: discriminability decreases monotonically as the transformations become more flexible (SoftMatch > Procrustes > Linear Predictivity). The constraints imposed by less flexible mappings appear to preserve family-specific signatures that are lost when arbitrary linear transformations are allowed.

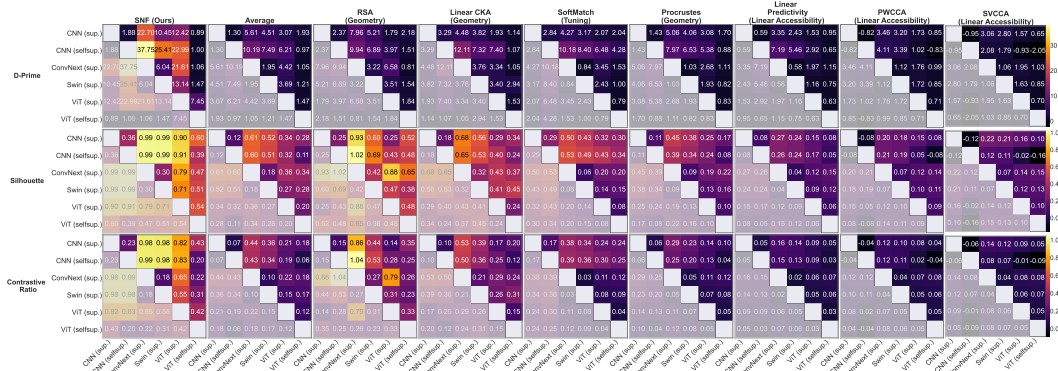

Figure 2: Model-family separability on ImageNet under $d'$, silhouette score and contrastive ratio. Columns correspond to nine similarity metrics, including two fusion-based methods (SNF, average) and seven commonly used representational metrics (Distinct aspects of representation emphasized by each metric are shown in the bracketed text). Fusion-based metrics consistently yield higher scores, highlighting their effectiveness in capturing family-level distinctions.

In contrast, metrics capturing linearly accessible information show substantially weaker discrimination between families. Linear Predictivity demonstrates the lowest separability among direct mapping-based metrics ($d' = 2.09$, silhouette = 0.14), while CCA-based metrics (PWCCA: $d' = 1.55$; SVCCA: $d' = 1.02$) show even weaker family separation. The weak discrimination of CCA-based metrics is particularly revealing. CCA identifies maximally correlated linear projections between representations, finding shared subspaces that are invariant to invertible linear transformations. CCA loads on the linear-accessibility facet: it detects shared linearly decodable subspaces but, unlike RSA/CKA or Procrustes/SoftMatch, it does not constrain or preserve representational geometry or tuning. The invariance of CCA to linear transformations, which makes it powerful for finding shared structure across superficially dissimilar representations, also makes it insensitive to the geometric and topological features that distinguish model families.

The theoretical relationships among these metrics help explain the discrimination hierarchy. RSA and Linear CKA are mathematically equivalent under appropriate centering (Williams, 2024) and both preserve the geometric structure of representations—they compare how similarly models organize their representation spaces without fitting any transformation. In contrast, the mapping-based

metrics show decreasing discrimination as they allow increasingly flexible transformations: Soft-Match permits only permutations that preserve individual unit correspondences, Procrustes allows orthogonal transformations (rotations and reflections), while CCA searches for optimal linear projections that maximize correlation. Linear Predictivity provides the most flexibility, allowing any linear transformation that minimizes prediction error.

The weak discrimination of metrics that assess linearly accessible information might suggest that this aspect of representation is more consistent across model families than geometric organization. Whether this reflects convergent computational strategies, methodological limitations of these metrics, or task-imposed constraints remains an open question.

## 3.2 INTEGRATION ACHIEVES SUPERIOR MODEL FAMILY DISCRIMINATION

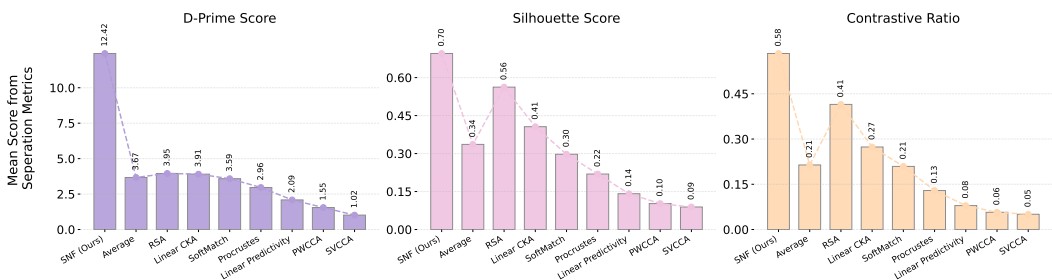

Figure 3: Mean model-family separability on ImageNet, evaluated using $d'$, silhouette score, and contrastive ratio. Fusion-based metrics (SNF, Average) outperform individual similarity metrics across all datasets, with SNF yielding the most consistent and robust separation. Scores are shown in their native scales and are not directly comparable across measures.

Critically, our SNF approach, which integrates information across all representational dimensions, achieves dramatically superior family separation compared to any single metric. SNF attains a $d'$ of 11.84—nearly three times higher than the best-performing single measure—and consistently outperforms all baselines across separation criteria. Importantly, as shown in Figure 2, SNF maintains high and balanced discrimination across nearly all family pairs. By contrast, individual metrics often exhibit uneven performance, separating some families while failing or for others.

Averaging similarities across metrics does not resolve this limitation: simple means dilute complementary signals and retain conflicting noise. In contrast, SNF's diffusion-based fusion reinforces consistent neighborhood structure across metrics while attenuating discordant components, yielding both stronger global separation and greater local stability.

This superior performance demonstrates that different representational dimensions provide complementary information about model families. While geometry and tuning capture family-specific computational strategies, and linearly accessible features potentially reflect more universal solutions, the integration of these diverse facets yields comprehensive signatures that most reliably distinguish model families.

To test whether SNF recovers the shared structure across metrics rather than replicating any single one, we try to quantify the intermetric agreement. Specifically, to ensure comparability across metrics, we symmetrize every similarity matrix by averaging it with its transpose, remove the diagonal (self-similarity), vectorize the remaining upper-triangle entries, and then compute correlation between vectors. As shown in Figure E.1, geometry-preserving metrics (RSA, SoftMatch, CKA) show strong mutual agreement, mapping-based metrics (Procrustes, Linear Predictivity) are moderately aligned, and CCA-variants highly correlate with each other but less agree with other metrics. SNF aligns only moderately with any single metric and is clearly distinct from simple averaging, indicating it fuses complementary facets instead of collapsing to one metric.

To further validate the SNF-derived similarity structure, we incorporate analyses based on biological neural data (Fig. I.1, I.2). Using anatomically ordered visual areas across subjects, we find that SNF more accurately recovers expected relationships such as the alignment of homologous regions across subjects and the established ventral-stream hierarchy than any individual metric.

Having shown that integrated representational signatures provide the most comprehensive characterization of model families, we next use the SNF-fused similarity matrix to derive a data-driven typology. This typology reveals how models cluster according to their representational processing, moving beyond surface-level groupings defined by architecture or training paradigm.

As a baseline, we first examine clustering results from individual metrics (Figure 4, F.2, F.4, F.6). We observe distinct patterns across the metrics: measures such as Linear Predictivity and Procrustes, tend to produce relatively uniform similarity values across a wide range of models, resulting in diffuse, non-distinct clusters. SoftMatch, which emphasizes the geometric alignment of individual units, struggles to clearly separate models that are neither CNN-based nor supervised. PWCCA and SVCCA produce noisy similarity matrices in which, despite dendrogram reordering, no strong or coherent clustering structure is apparent. RSA reveals some clustering—most notably a separation between CNNs and Transformers—but the partitions appear quite diffuse. Even simple averaging of normalized metric values fails to produce sharply defined clusters. Together, these results highlight the limitations of single metrics: each emphasizes a different facet of representational similarity, leading to clustering patterns that are fragmented, noisy, or inconsistent across metrics.

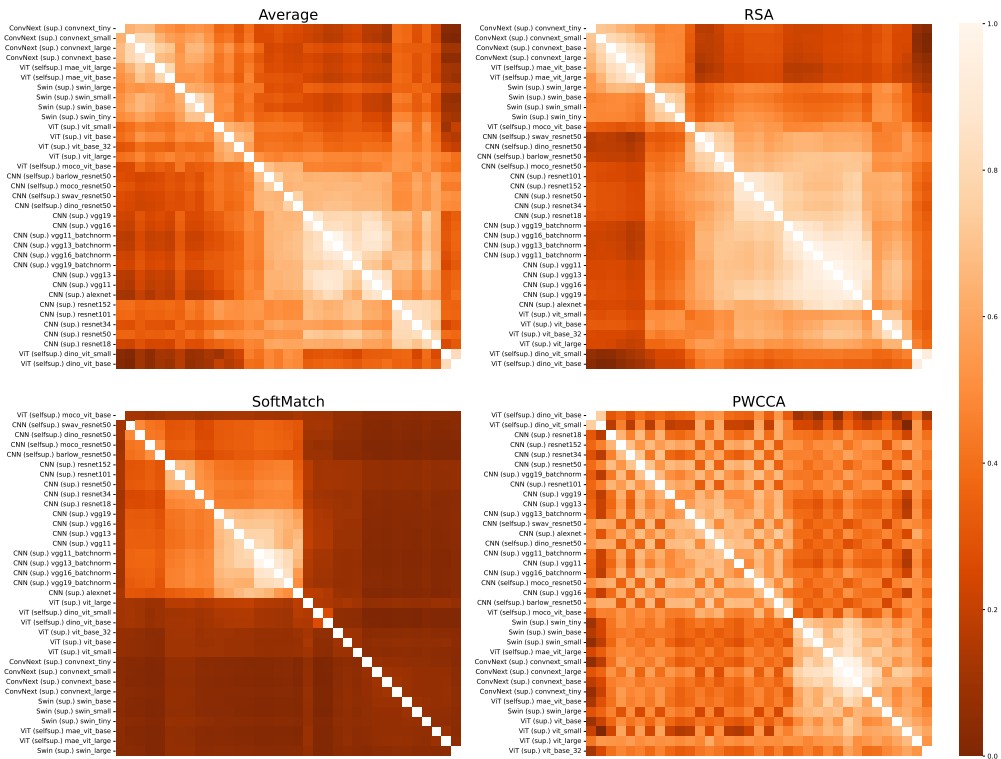

Figure 4: Hierarchical clustering of models using three functionally distinct representational similarity metrics and a similarity-metric–averaging baseline (see Fig. F.1 for additional metrics). Clustering is performed with average linkage and optimal leaf ordering, based on induced distances (1− similarity score). Rows and columns are deliberately re-ordered to match the leaf ordering produced by the clustering algorithm. Lighter colors indicate higher similarity; diagonal entries (self-comparisons) are omitted.

In contrast, hierarchical clustering of the fused similarity matrix reveals well-defined groupings that both confirm expected relationships and uncover surprising organizational principles (Figure 5, F.3, F.5, F.7)). To quantitatively assess the fidelity of these clusters, we compute the cophenetic correlation coefficient (CCC) measuring how well the clusters preserves the original pairwise similarities (Figure G.1). Our SNF-based metric achieves one of the highest CCC of 0.982 among all metrics indicating that the clusters formed by the fusion matrix more faithfully reflect the underlying similarity structure among models. These results provide additional evidence that the SNF produces more

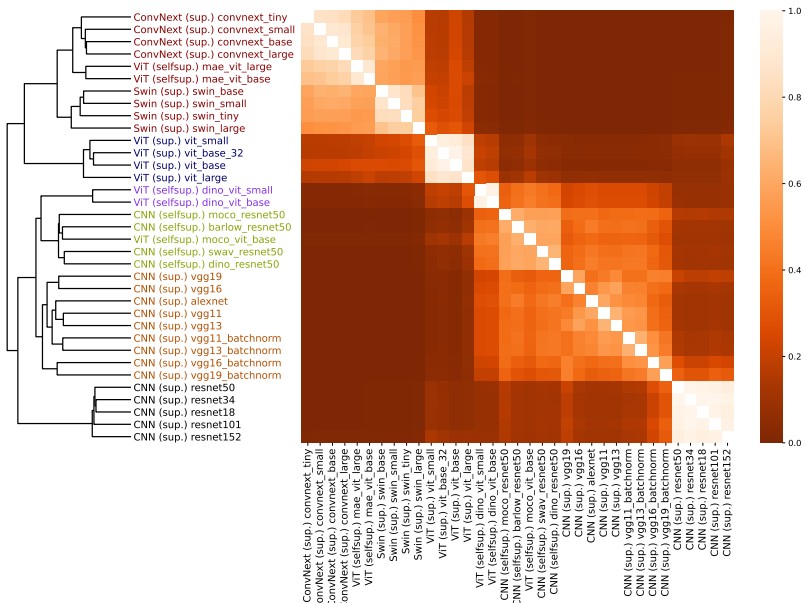

Figure 5: SNF-based clustering reveals that models naturally group by architecture and supervision regime. Supervised CNNs and ViTs each form distinct clusters; hybrid models (ConvNeXt, Swin) and MAE ViTs cluster together; and self-supervised models (e.g., DINO ViTs, self-supervised CNNs) form coherent groups. The heatmap shows the SNF-fused similarity matrix reordered by leaf ordering. Leaf labels are colored by the cluster (formed by SNF) they belong to; dendrogram cuts yield up to six flat clusters aligned with canonical categories.

meaningful and robust clusterings compared to individual baseline metrics, providing confidence in the typology.

Next, we examinine the clusters derived from the fusion matrix (Figure 5). Our typology reveals a complex interplay between architecture and training paradigm in determining representational structure. While some clusters align with architectural expectations—supervised ResNets group together, as do supervised VGGs and supervised ViTs—the most striking finding is that training paradigm can override architectural boundaries. All self-supervised models, regardless of architecture, form a unified cluster that transcends the CNN-Transformer divide. Self-supervised ResNets (MoCo, DINO, SwAV, Barlow Twins) group more closely with self-supervised ViTs than with their supervised architectural siblings. This suggests that the computational strategies induced by self-supervised learning—whether through contrastive learning, self-distillation, or redundancy reduction—create a shared representational signature that dominates over architectural differences.

Perhaps most surprisingly, hybrid architectures (ConvNeXt and Swin) cluster with MAE models despite their different design philosophies. ConvNeXts modernize CNNs with Transformer-inspired components, Swins introduce CNN-like biases into Transformers, and MAE employs masked reconstruction—yet they converge on similar representational structures. This convergence suggests that architectural modernization and masked reconstruction, though approaching from different angles, arrive at similar computational solutions. These unexpected groupings demonstrate the power of our data-driven approach: we discover that emergent computational strategies—shaped by the interplay of architecture, training objective, and task demands—determine representational structure. The typology thus reveals the "species" of vision models, defined not just by their surface characteristics but by how they fundamentally process and organize visual information.

## 4 DISCUSSION

In this work, we systematically compared multiple representational properties across vision models—including geometric organization, tuning of individual neurons, and linearly accessible fea-

tures—to identify which aspects are universally shared versus model-family specific. Our analysis reveals that geometry-preserving or tuning-preserving metrics (RSA, Soft Matching) strongly discriminate between model families, while Linear Predictivity and CCA-variants shows weaker separation. By applying Similarity Network Fusion from multi-omics analysis, we integrated these diverse metrics, achieving superior separation of model families compared to any single metric. Moreover, hierarchical clustering on the SNF-integrated similarity matrix revealed a data-driven typology of vision models that transcends traditional architectural categories. Just as biological typologies require multiple markers, understanding model representations benefits from integrating complementary perspectives rather than relying on single metrics that capture only partial aspects of representational structure.

We note that SNF is not intended to replace individual similarity metrics, but to provide a stable consensus structure when those metrics disagree. The fused graph captures global, multi-metric regularities in representational organization which could be useful for identifying broad model families, or stable clusters. For mechanistic or fine-grained analysis, researchers can and should examine the individual metrics contributing to the SNF fusion to understand which invariances (e.g., geometric, tuning-curve, subspace) drive a specific similarity. In this way, SNF acts as a robust starting point, with the individual metrics supplying interpretability.

Our analysis has several important limitations. First, we focus primarily on natural image databases (ImageNet, Ecoset, CIFAR) and testing the stability of the discovered typology to more out-of-distribution domains would test the generalizability of these groupings. Second, we focus largely on penultimate layer representations. Though we observe reasonable consistency across layers for most metrics as shown in Figure H.1, comprehensive multi-layer analysis could reveal how representational strategies correspond across network depth. Third, a limitation is that all models used in this study were trained on ImageNet-1k. Exploring how the typology changes for web-scale SSL or image–text models would be an important direction for future work. Finally, while SNF provides a principled integration framework, our typology inherently depends on the choice of input metrics. Future work should systematically evaluate how metric selection influences the discovered groupings and explore alternative integration approaches to validate the robustness of our findings.

This typology framework opens several research avenues. Longitudinal analysis could track how models move through representational space during training, potentially revealing whether all models traverse similar developmental trajectories. Extending the framework to multi-modal models could test whether vision-language models form distinct clusters or integrate into existing groupings. A particularly important direction is comparing our representation-based typology with behavioral groupings. Our research focuses on representational similarity, whereas many "practical implications" of such analyses are actually closer to behavioral/functional similarity, which is another type of model similarity metric. The relationship between behavioral similarity and representational similarity requires further investigation and has been proposed as a complementary desideratum for evaluating representational similarity metrics (Ding et al., 2021; Bo et al., 2024). Do models that cluster together based on internal representations also exhibit similar patterns of errors, biases, or generalization behaviors? If behavioral measures yield different groupings than our SNF-based approach, this would reveal that representational similarity doesn't necessarily imply functional similarity. Finally, validating whether models within the same representational cluster show similar transfer learning performance (e.g., similar fine-tuning convergence rates or final accuracies on downstream tasks) could provide practical utility for the typology and guide model selection strategies. Our framework provides a tool for the community to assess whether new models offer genuinely novel representational strategies or represent variations on established themes.

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

## A  EXPERIMENT SETTINGS

**Datasets.**   The chosen datasets are balanced across different classes as shown in Tabel A.1.

Table A.1: Per-class and total sample counts for standard evaluation splits.

| Dataset | Number of Classes | Samples / Class | Total Samples |
|---|---|---|---|
| ImageNet-1k (valid) | 1,000 | 50 | 50,000 |
| Ecoset (valid) | 565 | 50 | 28,250 |
| CIFAR-10 (test) | 10 | 1,000 | 10,000 |
| CIFAR-100 (test) | 100 | 100 | 10,000 |

**Models.**   All models are trained on the ImageNet-1K training set. We obtain pretrained weights from torchvision (maintainers & contributors, 2016), Torch Hub (Paszke et al., 2019), timm (Wightman, 2019), or the official repositories. Unless noted otherwise, we extract activations from each model's penultimate layer. For CNNs, which commonly include global average pooling, we use that pooled feature. For ViT-style models, we average non-CLS token embeddings to form the final representation for consistency across architectures.

## B  MODEL FAMILY AND ARCHITECTURE CHOICES

We evaluate multiple architectures within each family to capture variation in depth, width, parameter amounts and design choices.

**Convolutional Neural Network (supervised; CNN (sup.)).**   Bottom-up hierarchies with convolutions and pooling that impose strong local inductive biases. We include AlexNet (Krizhevsky et al.), VGG-11/13/16/19 (with/without batch normalization) (Simonyan & Zisserman), and ResNet-18/34/50/101/152 (He et al.).

**Transformer (supervised; Trans (sup.)).**   Vision Transformers partition images into fixed-size patches and use multi-head self-attention for global interactions (Dosovitskiy et al., 2021). We include ViT-S/16, ViT-B/16, ViT-L/16, and ViT-B/32.

**ConvNeXt (Liu et al., 2022).**   A convolutional family inspired by Transformer design (e.g., large-kernel depthwise convolutions, patchified stems, inverted bottlenecks). We use ConvNeXt-Tiny/Small/Base/Large.

**Swin Transformer (Liu et al., 2021).**   A hierarchical Transformer with shifted window attention for efficient locality while retaining global context. We use Swin-Tiny/Small/Base/Large.

**Convolutional Neural Network (self-supervised; CNN (selfsup.)).**   Methods trained without labels using CNN backbones (ResNet-50). We include MoCo (Chen* et al., 2021), DINO (Caron et al., 2021), SwAV (Caron et al., 2020), and Barlow Twins (Zbontar et al., 2021), spanning contrastive and non-contrastive paradigms (momentum contrast, self-distillation, online clustering, and redundancy reduction).

**Transformer (self-supervised; Trans (selfsup.)).**   Label-free training with Transformer backbones. We include DINO-ViT-Small/16 and DINO-ViT-Base/16  (Caron et al., 2021), MoCo-ViT-Base/16 (Chen* et al., 2021), and MAE-ViT-Base/16 and MAE-ViT-Large/16 (He et al., 2021).

# C   FINE-GRAINED SEPARATION PERFORMANCE ON OTHER DATASETS

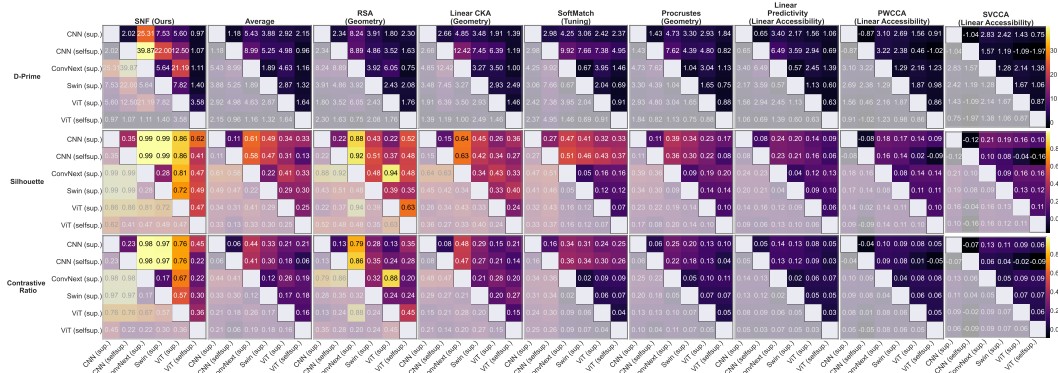

Figure C.1: Same as Figure 2, but using Ecoset instead of ImageNet.

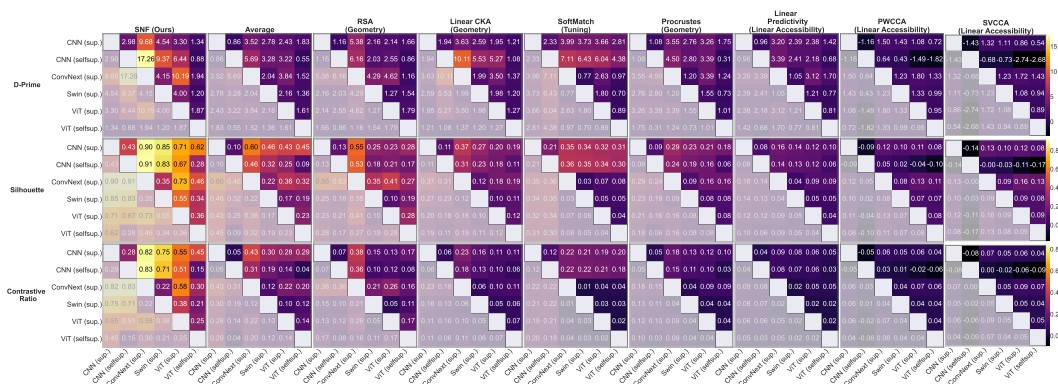

Figure C.2: Same as Figure 2, but using CIFAR10 instead of ImageNet.

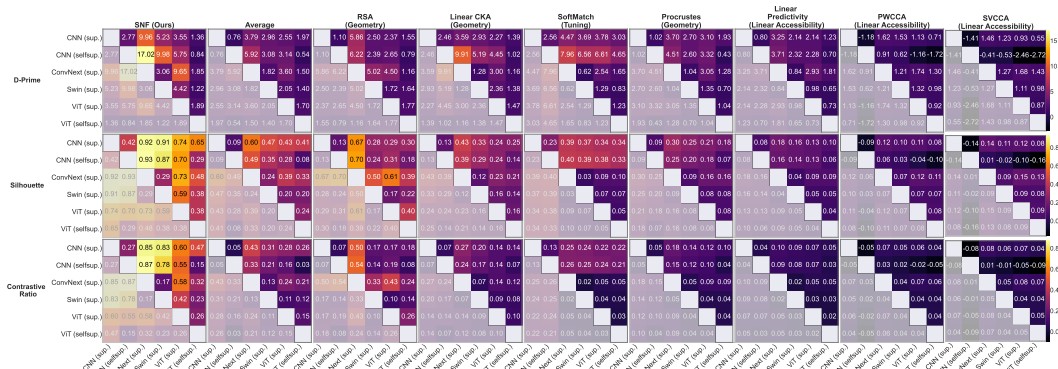

Figure C.3: Same as Figure 2, but using CIFAR100 instead of ImageNet.

# D    MEAN SEPARABILITY ON OTHER DATASETS

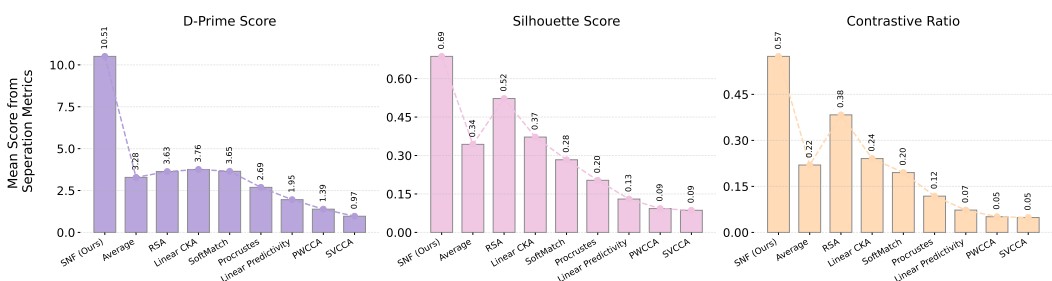

Figure D.1: Same as Figure 3, but using Ecoset instead of ImageNet.

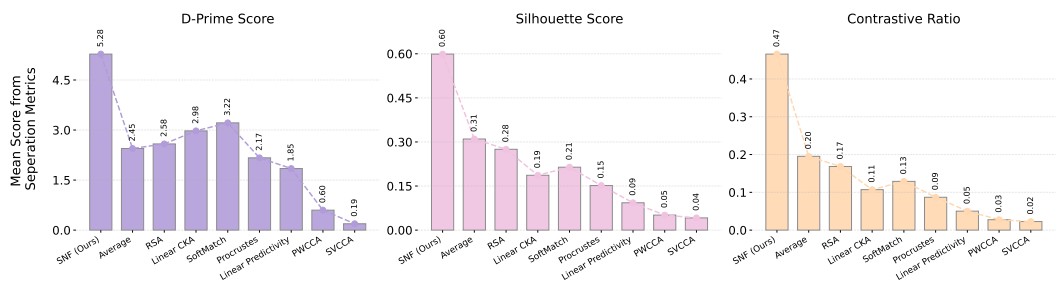

Figure D.2: Same as Figure 3, but using CIFAR10 instead of ImageNet.

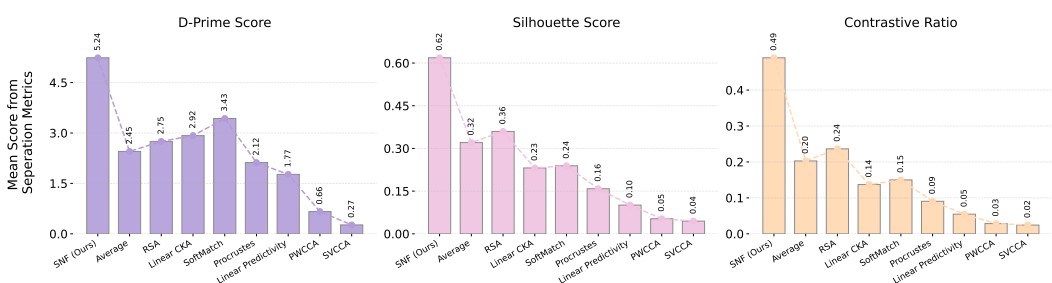

Figure D.3: Same as Figure 3, but using CIFAR100 instead of ImageNet.

# E  METRICS' SIMILARITY SCORES CONSISTENCY

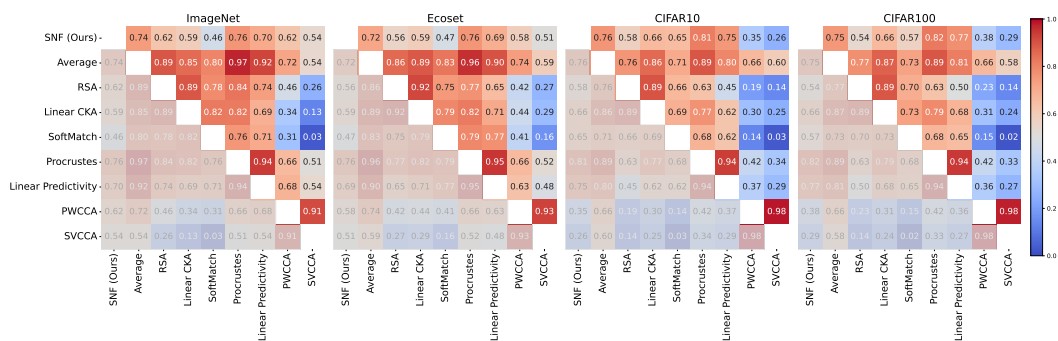

Figure E.1: Each subplot (ImageNet, Ecoset, CIFAR10, CIFAR100) shows pairwise Pearson correlations between the vectorized upper-triangle entries of the symmetrized model–model similarity matrices produced by nine metrics. Higher values indicate that two metrics have more similar relational geometry among models.

# F MORE CLUSTERING PERFORMANCE

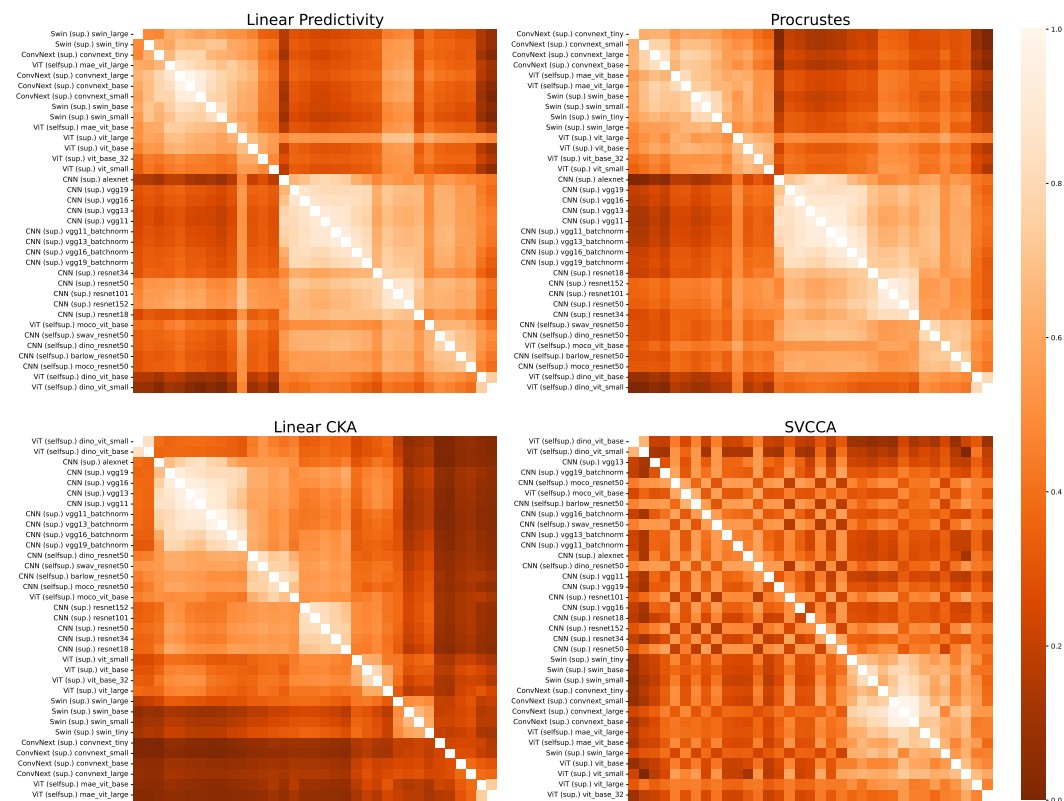

Figure F.1: Same as Figure 4, but for the other 4 metrics.

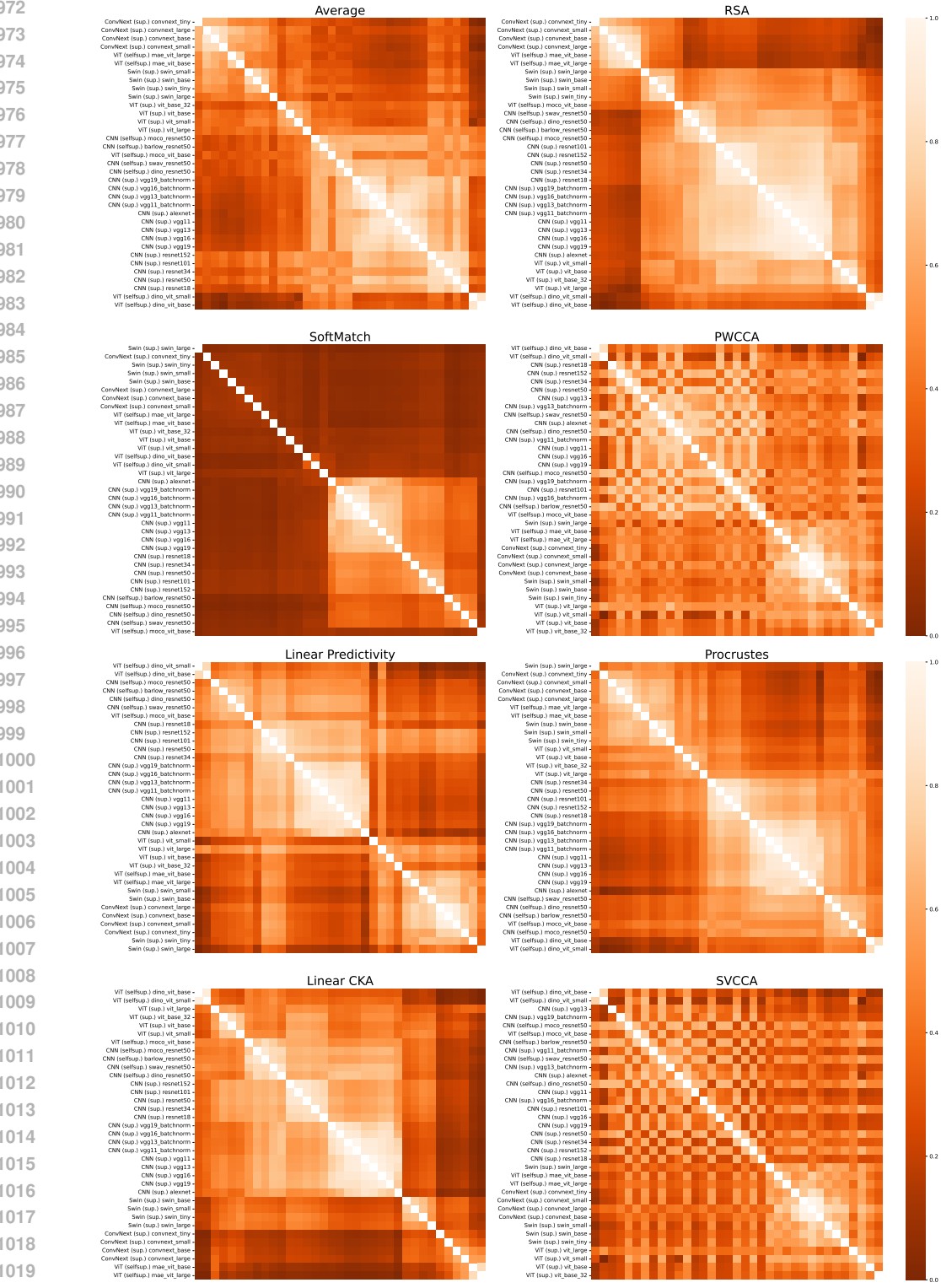

Figure F.2: Same as Figure 4 and Figure F.1, but using Ecoset instead of ImageNet.

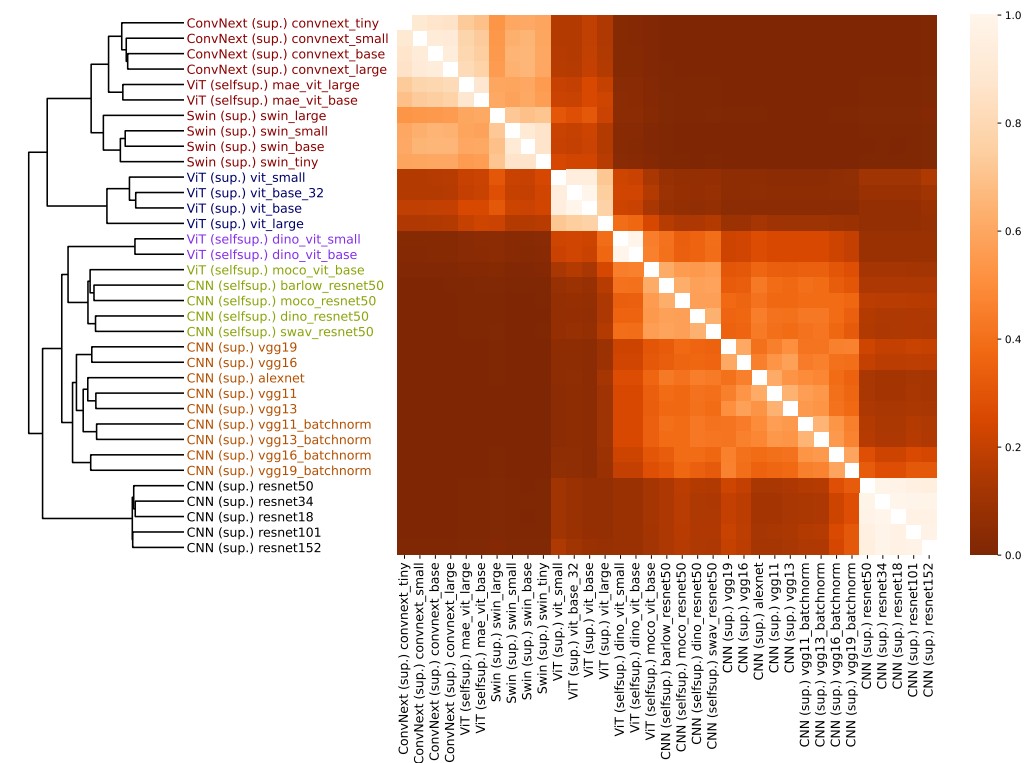

Figure F.3: Same as Figure 5, but using Ecoset instead of ImageNet.

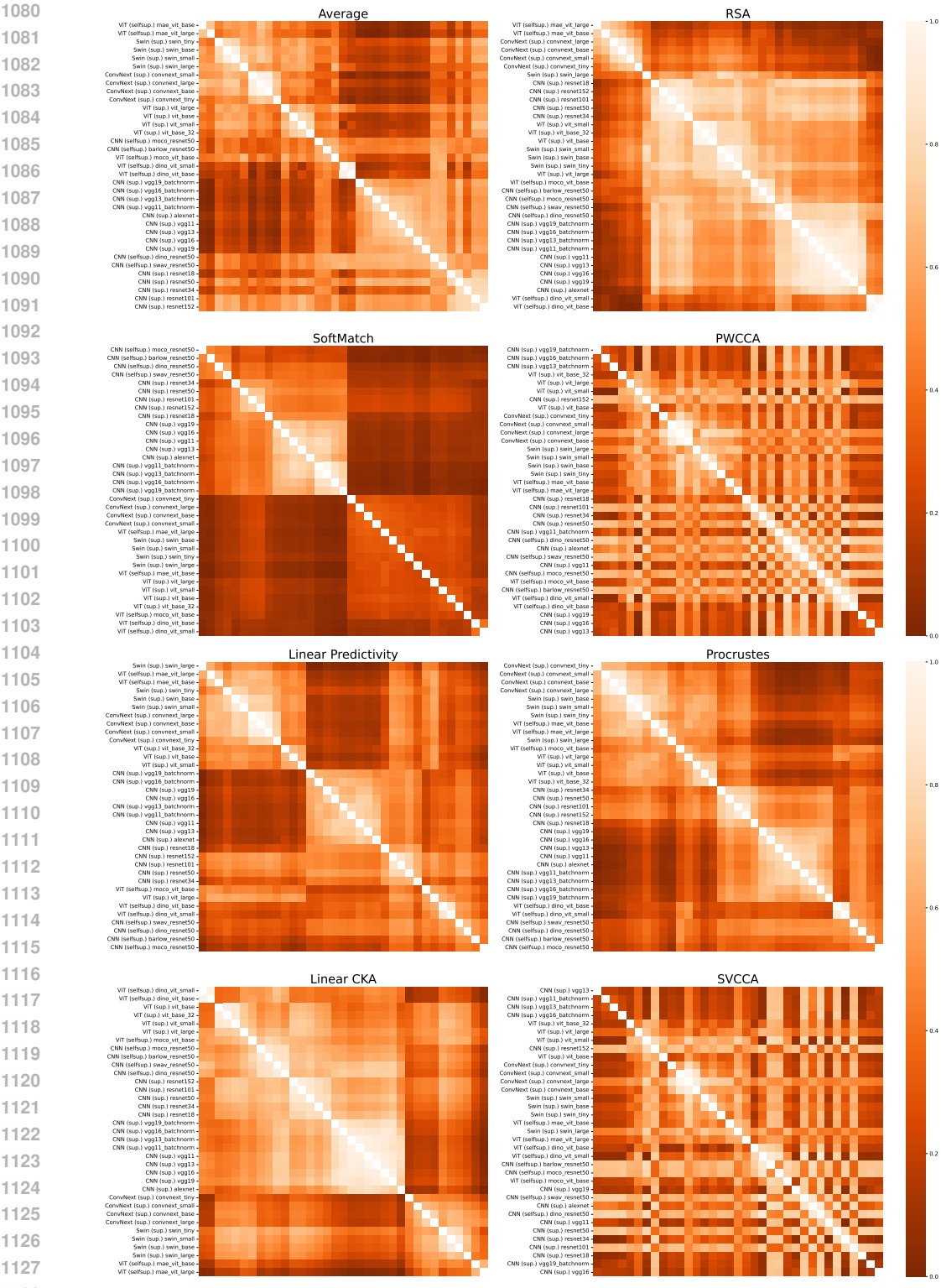

Figure F.4: Figure 4 and Figure F.1, but using CIFAR10 instead of ImageNet.

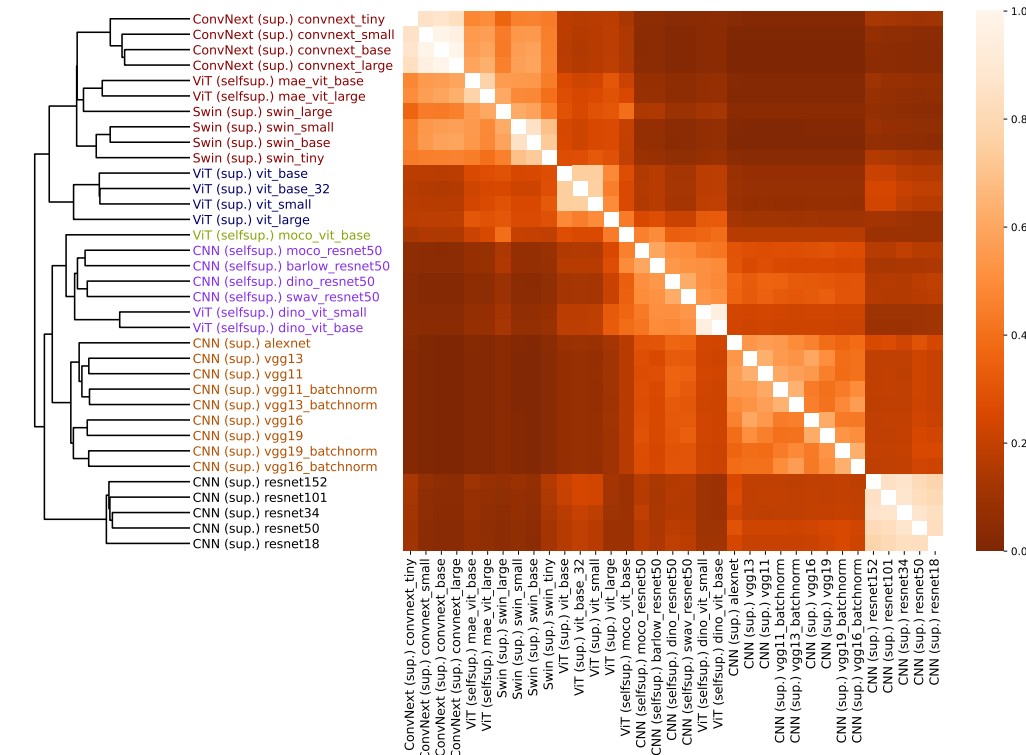

Figure F.5: Compared to ImageNet in Figure 5, the clustering result is a little different but similar, demonstrating the SNF results and metric's separability are also influenced by datasets, while also preserving a certain degree of stability. The supervised models are clustered in the same way, but the clusters for the self-supervised models changed, demonstrating that the self-supervised way leads to a special but kind of unifying representational geometry.

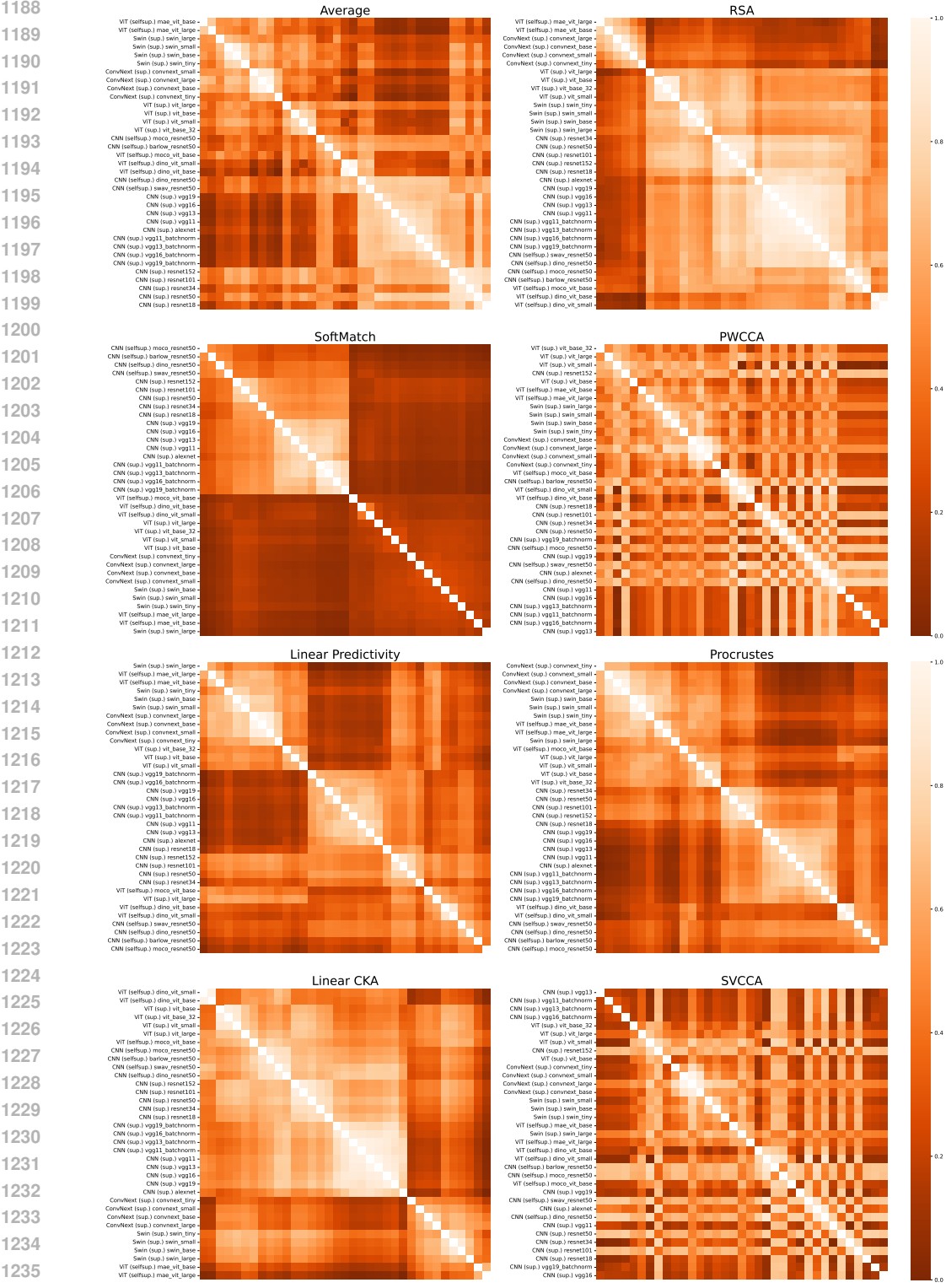

Figure F.6: Figure 4 and Figure F.1, but using CIFAR100 instead of ImageNet.

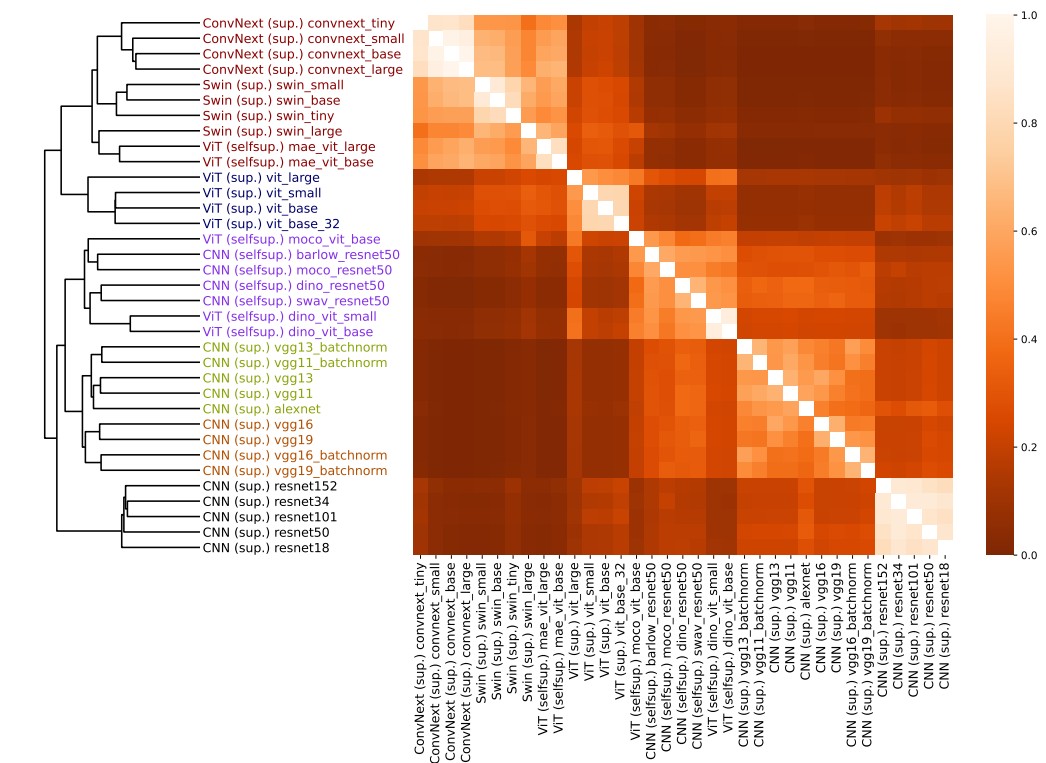

Figure F.7: Same as Figure F.5, but using CIFAR100 instead of CIFAR10.

# G  COPHENETIC CORRELATION COEFFICIENTS FOR CLUSTERING

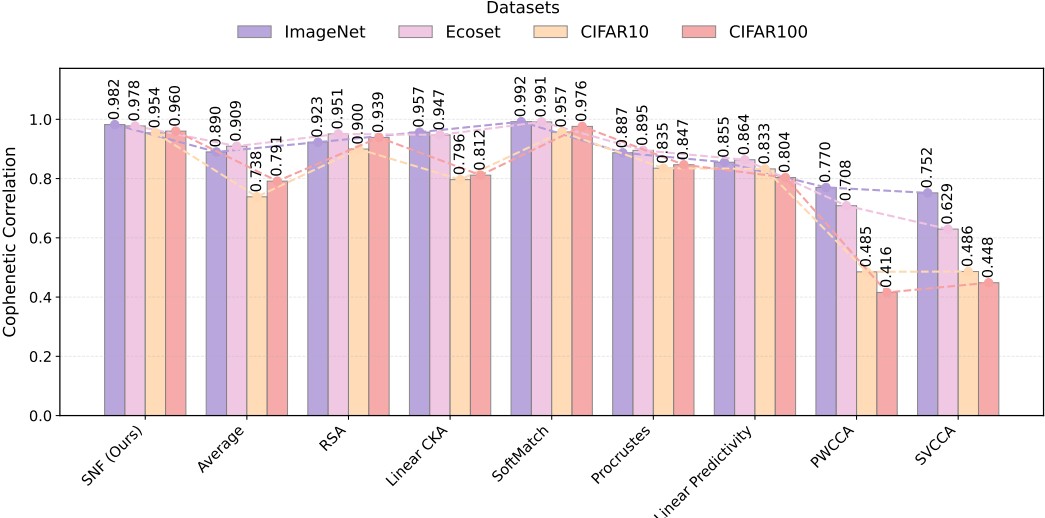

Figure G.1: Cophenetic correlation coefficients (CCC) for hierarchical clusterings induced by each metric on four datasets. Higher CCC (closer to 1) means the clustering more faithfully preserves the original pairwise structure. Bars show CCC on ImageNet, Ecoset, CIFAR10, and CIFAR100 (values annotated; lines trace dataset-wise trends).

# H    METRICS' CROSS LAYER CONSISTENCY

For each metric and dataset, we also test whether the similarity structure is stable across depth. For CNNs, if batch normalization layers exist (e.g. ResNet), we count layers by them; if not (e.g., VGG, AlexNet), we count layers by ReLU units. For ViTs, one feature extraction unit (first layer normalization + attention block + second layer normalization) is counted as one layer, and we take the output after the second layer normalization. We select the layer for a normalized depth $d \in (0, 1]$ via $\ell = \lfloor dL \rfloor$, where $L$ is the total number of layers. For each metric, one depth corresponds to one matrix. We vectorized the off-diagonal entries, and computed Pearson correlations between the 3 depth pairs. We found that SNF and SoftMatch show the highest depth consistency, whereas the CKA and CCA variants are less stable. Although the representations in different layers could be greatly different, the SNF could still identify the layer-model belonging relationship and the difference between model families.

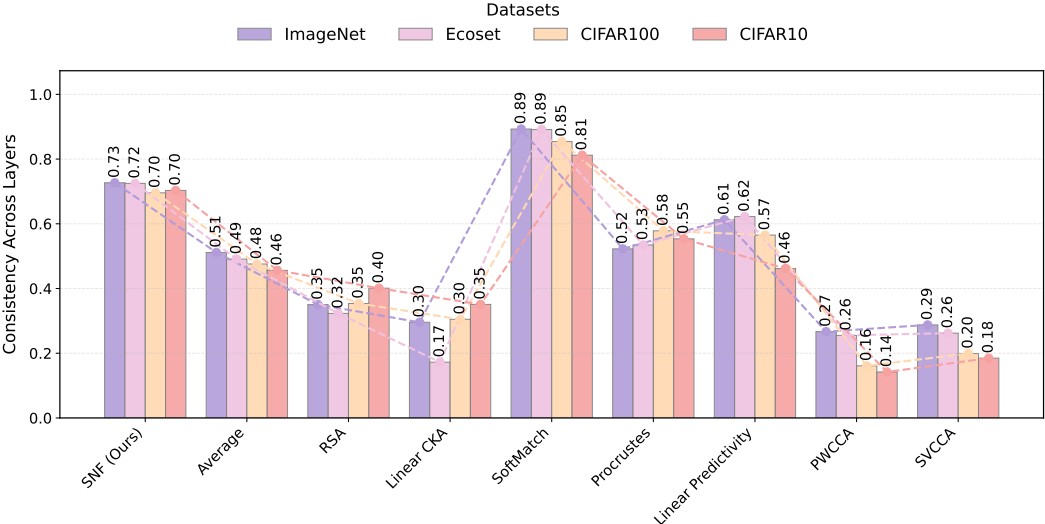

Figure H.1: Cross-layer consistency of inter-model similarity. The bar height is the mean of the three depth pairs correlations; higher values indicate greater consistency.

# I   BRAIN CLUSTERING VALIDATION

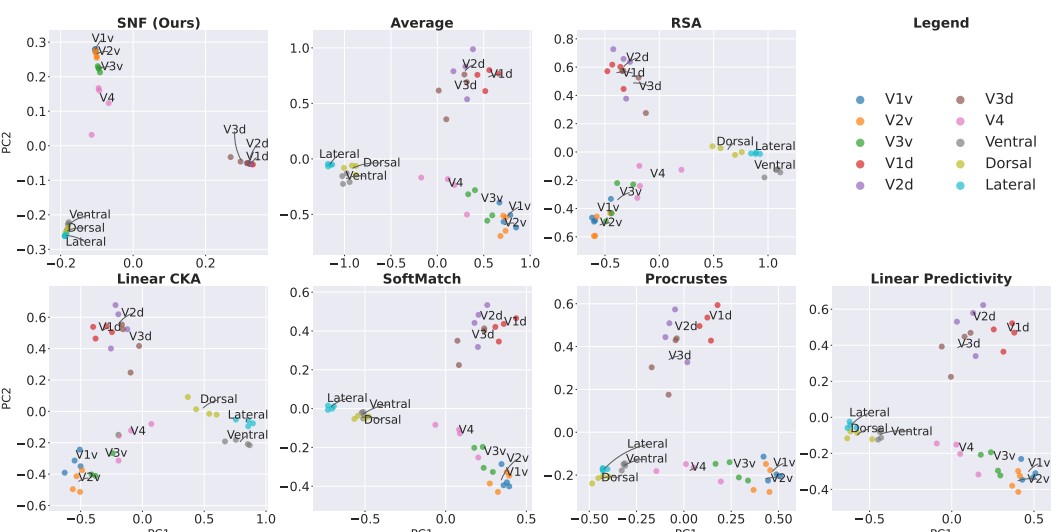

Figure I.1: Cross-regional relationships were derived from five similarity measures using PCA analysis on NSD data. Each point represents a brain region instance, and text labels indicate centroid positions. SNF fusion shows the best intra-class compactness and inter-class separation.

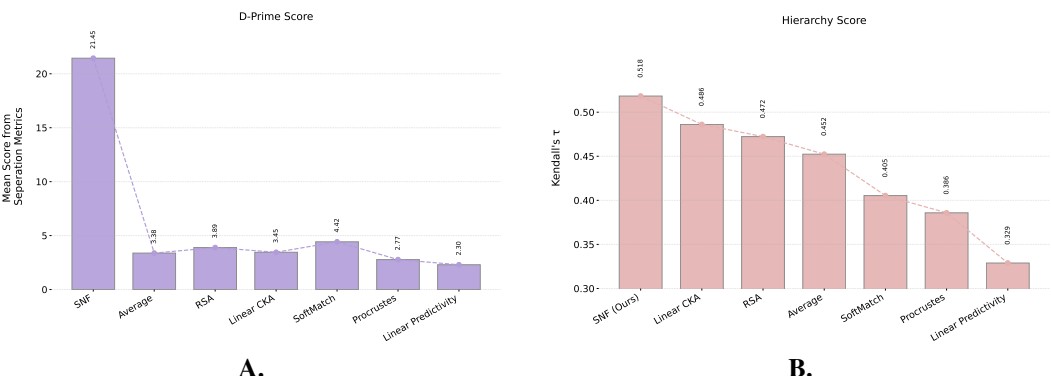

Figure I.2: **A.** Mean $d'$ score on NSD. **B.** Significant hierarchical scores comparison of different representations across the ventral visual pathway (NSD, $n = 1000$, $p < 0.05$).

For brain data, we used responses to the 1,000 shared natural images from the Natural Scenes Dataset (NSD) (Allen et al.), spanning 10 visual regions across 4 subjects. We exclude SVCCA and PWCCA from brain analyses because their alignment values were inflated by the high voxel count relative to the number of stimuli.

To assess brain region discriminability, we quantified the difference between within-region and across-region representational similarities using the separability measure $d'$. SNF which integrates information across all representational dimensions, achieves dramatically superior family separation compared to any single metric. In Figure I.2A, SNF attains a mean $d'$ of 21.45—nearly five times higher than the best-performing single measure—and consistently outperforms all baselines across separation criteria. To visualize and corroborate these patterns, we applied PCA to each similarity matrix. All metrics recovered broad trends (Figure I.1), but the SNF-fused matrix produced the clearest, most interpretable layout. Early visual areas (V1-V4) traced a smooth trajectory, with V4 in a transitional position between the early cortex and higher-level ventral, dorsal, and lateral streams.

To quantify how well a representation preserves the expected ventral-stream hierarchy, as suggested in Thobani et al. (2025), we use a hierarchy score based on Kendall's $\tau$. Each region $r$ is assigned

a discrete hierarchical level $L(r) \in \{1, \ldots, 5\}$, as $L(\text{V1v}) = L(\text{V1d}) = 1, L(\text{V2v}) = L(\text{V2d}) = 2, L(\text{V3v}) = L(\text{V3d}) = 3, L(\text{V4}) = 4, L(\text{ventral\_visual}) = 5$, given a brain-region-by-region similarity matrix $S \in \mathbb{R}^{R \times R}$ and the corresponding region labels $\{r_1, \ldots, r_R\}$, we first construct a hierarchy-distance matrix,

$$H_{ij} = \bigl| L(r_i) - L(r_j) \bigr|, \quad 1 \le i, j \le R.$$

We then extract all off-diagonal entries of $H$ and $S$,

$$\mathbf{h} = \{H_{ij} : i \ne j\}, \qquad \mathbf{s} = \{S_{ij} : i \ne j\},$$

and define $\mathbf{y} = -\mathbf{s}$ so that smaller hierarchy distance corresponds to larger similarity. The hierarchy score is the Kendall's $\tau$-b rank correlation between $\mathbf{h}$ and $\mathbf{y}$: $\tau_{\text{hier}} = \tau_{\text{Kendall}}(\mathbf{h}, \mathbf{y})$. These results further support the validity of SNF in revealing known biological correspondence structure (Figure I.2B).

## J  THE USAGE OF LARGE LANGUAGE MODELS (LLMS)

LLMs are mainly used in two ways. For aiding or polishing writing, they are primarily used to identify typos and make the language more aligned with conventions of academic writing. For retrieval and discovery, LLMs with internet access are used to search for related work.

