# OpenReview forum: "A Data-driven Typology of Vision Models from Integrated Representational Metrics"
_ICLR.cc/2026/Conference — Submitted to ICLR 2026_

### Official Review · Reviewer_tjUP · 2025-10-29

**Soundness:** 2
**Presentation:** 2
**Contribution:** 2
**Rating:** 2
**Confidence:** 4

**Summary:**

This paper presents a comparison of ImageNet-pretrained computer vision models using multiple representational similarity metrics (RSA, CKA, SoftMatch, etc.) and adapts Similarity Network Fusion (SNF) from multi-omics biology to integrate these metrics. Every representational similarity metric has its own characteristics and invariances, making similarity assessments metric-dependent. To evaluate metric differences, the authors group 35 vision models into four primary "model-families" (supervised CNNs, self-supervised CNNs, supervised Vision Transformers, self-supervised Vision Transformers) plus two CNN/ViT hybrids (ConvNeXt, Swin). They use model-family separability as a criterion to select the best metric, finding that geometry/tuning-preserving metrics discriminate better between families than metrics measuring "linearly accessible information" (metrics learning an affine transformation between the representations). SNF achieves the strongest separation by integrating multiple metrics. The authors then apply hierarchical clustering to the SNF-fused similarity matrix to discover a "data-driven typology," which mostly recovers the pre-defined families. The typology reveals that self-supervised models cluster together across architectural boundaries, and hybrid architectures (ConvNeXt, Swin) cluster with masked autoencoders (MAE).

**Strengths:**

1. The adaptation of Similarity Network Fusion (SNF) to integrate multiple representational similarity metrics is methodologically sound and addresses the limitation that individual metrics capture only partial aspects of representational structure. SNF achieves stronger family separation than any single metric alone, demonstrating the value of integration. This framework could be extended in interesting directions not explored in the paper, such as integrating local and global representational similarity (e.g., using different CKA kernels) or combining layer-wise and network-wide comparisons.

2. Cross-dataset robustness checks: Testing the typology across four datasets and showing consistent patterns (Figures 3, C.1-C.3) strengthens confidence that the findings reflect genuine representational properties rather than dataset-specific artifacts.

**Weaknesses:**

- Major
    1. Circular reasoning / "test-set tuning" problem: As far as I understand, the most critical point is using pre-defined model-family labels to evaluate which metrics best separate these families (Section 2.3, Figure 2), then showing that the integrated metric SNF is “winning “in this aspect and then using SNF to claim "data-driven" discovery of clusters that largely match the original labels. This is analogous to selecting a model based on test set performance.
    2. Limited validation of linear accessible information interpretation: The authors interpret weak discrimination by  linear accessible information metrics as evidence that models "converge on similar linearly accessible information" (L320-323). However, these metrics (Linear Predictivity, CCA variants) quantify similarity via correlation between the target representation and its linear reconstruction. Correlation may be insufficiently sensitive—high correlation can occur even when reconstructions are poor in absolute terms, potentially inflating similarity scores. The paper would benefit from: (a) complementary reconstruction quality metrics (R², mean squared error, variance explained) to verify that high correlation reflects genuine similarity rather than metric leniency, and/or (b) downstream task validation (linear probing, transfer learning) to test whether metric-based similarity predicts functional similarity. Without these checks, it's unclear whether weak discrimination reflects genuine convergence in linearly accessible information or limitations of correlation-based evaluation that fails to detect meaningful representational differences
- Minor
    1. Limited model scope undermines relevance: The authors restricted themselves to ImageNet-1k pretrained models to avoid training data domain to be a influencing factor on the representational similarity values. However,  restricting to ImageNet-1k pretraining excludes modern web-scale and multimodal encoders (CLIP, DINOv2, SigLIP, EVA-CLIP) that dominate current practice. The typology characterizes legacy models but may not generalize to the contemporary landscape. Self-supervised model diversity is particularly limited (mostly ResNet-50 variants). The claimed "data-driven typology" framework's value depends on its applicability to current models.
    2.  Inconsistent terminology and poor figure design:
         - Terms self-supervised vs unsupervised (label of the methods) used inconsistently.
         - It would have been easier to follow the information/ observations if the authors had thought of a method to aggregate the values in Figure 2 and putting the correct table
         - Figure 4's different orderings across subplots make cross-metric comparison impossible, and Figures 4-5 are difficult to read with labels too small.
         - Figure 2 should clearly indicate which metrics measure which facets (geometry/tuning/linear-accessibility) or alternatively using a table.
         - Unmotivated metric fusion: Section 2.4 jumps directly to SNF implementation without justifying why metric integration is needed. What problem does integration solve that single metrics don't? The paper would benefit from explicit motivation (e.g., showing that single metrics give contradictory groupings, or that different metrics capture complementary information that's individually incomplete).

**Questions:**

- Circular reasoning: How do you address the concern that using family labels to select the best metric (Section 2.3), then using that metric to discover families, constitutes circular reasoning?
- Functional validation: Can you provide downstream task performance (linear probing, transfer learning) to validate whether weak linear-accessibility discrimination reflects genuine convergence or just methodological artifacts? *This could be a lot of work*. In case limited time, could the authors simply comment on this concern?
- Pooling choice: Why use average pooling instead of the tokens models were trained with (e.g., CLS for DINO)? Did you test sensitivity to this choice?
- Modern models: Would the authors expect typology change if including web-scale models (CLIP, DINOv2)? Do you expect family boundaries to remain stable?
- Prior work: How does your approach differ from prior model clustering work (Kornblith 2019, Ding 2021, Rosenfeld 2021)? Is SNF integration the “only” novelty?
- Figure 4 patterns: Why do VGG models show less within-family similarity than ResNets in most metrics, despite both being supervised CNNs?

---

> ### Author Response · Authors · 2025-11-22
>
> ## R4-W-Major 1: Circular Reasoning / "Test-Set Tuning" Concerns
> We appreciate the reviewer’s careful attention to the possibility of circularity. However, our use of model-family labels differs fundamentally from “test-set tuning” in one key respect: SNF is fully unsupervised. It never sees, uses, or is adjusted based on family labels. No parameter in SNF is selected, tuned, or weighted using family separability. It simply fuses multiple similarity matrices in a hypothesis-agnostic manner.
>
> To clarify the workflow, we first compare individual similarity metrics (RSA, Procrustes, Linear Predictivity, SoftMatch, etc.) against family labels only as an external diagnostic which serves as a pragmatic sanity check that a metric can at least distinguish broad architectural/training-paradigm differences such as ResNet vs ViT or supervised vs contrastive learning. Importantly, this step is not used to select hyperparameters for SNF or to modify the fusion procedure in any way and is done solely to confirm that metric aggregation via SNF is capable of giving meaningful results. After evaluating metrics, the SNF procedure, whose construction and parameters are fully fixed, is applied to fuse the metrics. The resulting clustering is then compared to family labels after the fact, simply to assess whether the unsupervised structure aligns with well-known distinctions. This is analogous to using labels to interpret unsupervised clusters, not to train them.
>
> Thus, the family labels function solely as an external reference, not a “test set” used during model or parameter selection. No part of the SNF algorithm is optimized using those labels, making the circularity concern inapplicable.
>
> We also agree with the deeper point underlying the reviewer’s concern: there is no true ground-truth taxonomy for DNNs. A central theme of our work is that the absence of such a gold standard makes it inherently difficult to evaluate representational similarity metrics. Our use of family labels is therefore a pragmatic proxy for gross structural differences and we use this to validate metrics.
> To address this limitation more rigorously, we introduce a second, independent form of validation based on biological neural data (Appendix I). Unlike artificial model families, the organization of the primate visual system is well-established: early visual areas (V1, V2, V3, V4) form a progression of increasing complexity, and higher-level ventral regions (e.g., ventral visual) sit further along this hierarchy. Moreover, homologous regions (e.g., V1 across different subjects) are expected to exhibit similar representational structure. These properties have been demonstrated consistently across decades of neurophysiology and fMRI research, and thus provide a robust external benchmark.
>
> Using responses from 10 visual regions in the Natural Scenes Dataset (NSD), we evaluate each similarity metric on their ability to recover this canonical hierarchy and to distinguish homologous from non-homologous regions. Across all separability criteria (d′, PCA embedding, and a Kendall’s τ hierarchy score), SNF substantially outperforms all individual metrics. For example, SNF achieves a mean d′ of 21.45 (nearly 5× the next best metric) and yields the clearest early-to-late cortical trajectory in PCA space. The fused similarity matrix recovers both (i) tight clustering of the same region across subjects and (ii) the expected order along the ventral stream—patterns that individual metrics recover only partially or not at all.
>
> These biological results strengthen confidence that SNF captures meaningful structure independent of any architecture or supervision labels, and that the taxonomy it produces is not an artifact of circular reasoning.

---

> ### Author Response · Authors · 2025-11-22
>
> ## R4-W-Major 2: Verification of Linear Accessible Information.
>
> We thank the reviewer for raising this important point. We agree that correlation-based linear-accessible-information metrics (e.g., Linear Predictivity, CCA variants) can exhibit insensitivity in certain regimes, and that high correlation values do not necessarily imply high alignment. In the current version of the paper, our interpretation of these results is already intentionally cautious: we explicitly state that “weak discrimination of metrics that assess linearly accessible information might suggest that this aspect of representation is more consistent across model families than geometric organization. Whether this reflects convergent computational strategies, methodological limitations of these metrics, or task-imposed constraints remains an open question.”
>
> To avoid any over-interpretation, we have further tempered this language in the revision. Our goal is not to claim that models genuinely converge in linearly accessible information, but rather to highlight that these specific metrics provide limited discriminability.
>
> We agree that incorporating behavior similarity (linear probing, transfer learning) would be highly valuable; however, this is beyond the scope of this paper which focuses on comparing similarity metrics themselves and constructing a stable, aggregated representational topology. We shall explicitly designate this as a significant open question in the discussion, rather than over-extending our current findings.

---

> ### Author Response · Authors · 2025-11-22
>
> ## R4-W-Minor 1: Limited model scope (lacking web-scale/multimodal encoders)
> We appreciate the reviewer’s concern regarding the scope of models included. We agree that incorporating contemporary web-scale and multimodal encoders (e.g., CLIP, DINOv2, EVA-CLIP, SigLIP) is an important next step, and we now make this explicit in the Discussion.
>
> In this paper, we deliberately restricted the model pool to encoders pre-trained on ImageNet-1k. The primary aim was to compare the impact of different model families (defined by training objectives and architectures) on representations within a unified data and label space, thereby avoiding training data distribution itself becoming major confounding factors. Recent work (e.g., Conwell et al., 2024) shows that “visual diet” can be the main driver of representational differences, often overshadowing architectural or objective-level factors, and our design follows that observation. It is precisely this controlled setting that allowed us to observe that, under identical data conditions, SSL models cluster together across architectures. Incorporating web-scale SSL or image-text models such as DINOv2/CLIP directly would simultaneously alter both training objectives and data distribution, rendering the clustering structure harder to interpret: what exactly drives the differences—the data, the objectives, or the architecture? In the revised version, we have articulated the rationale behind this experimental setup more clearly (in the methods section; highlighted in red) and designated extending it to web-scale/multimodal models as a significant direction for future work (see the Discussion).
>
> Finally, we emphasize that the contribution of our work is not the specific taxonomy, which will naturally evolve as new model families emerge. The contribution is the methodological framework: a robust, data-fusion–based approach (SNF) for constructing stable, interpretable representational organization structures that are less sensitive to any single metric.
>
> Refs:
>
> Conwell, C., Prince, J. S., Kay, K. N., Alvarez, G. A., & Konkle, T. (2024).
> A large-scale examination of inductive biases shaping high-level visual representation in brains and machines. Nature Communications, 15, 9383.

---

> ### Author Response · Authors · 2025-11-22
>
> ## R4-W-Minor 2: Inconsistent terminology, graphic design issues
>
> We appreciate these specific suggestions.
>
> - We have consistently used the term "self-supervised" throughout the paper to avoid mixing "self-supervised" and "unsupervised."
>
> - The aggregation of the values in Figure 2 is shown in Figure 3 using averaging.
>
> - For Figure 4: In hierarchical clustering heatmaps, the row/column ordering is part of the output of the algorithm: re-ordering rows and columns according to the dendrogram is precisely what reveals a metric’s clustering structure. Using a fixed ordering across metrics (e.g., based on one metric’s dendrogram) would artificially constrain the others and obscure their intrinsic structure, making it impossible to see whether a metric produces coherent families or collapses them.
>
> - Considering the figure sizes, we have relocated some subplots in Figure 4 to the appendix to free space for enlarging key information in the main text and enlarged the size of Figure 5.
>
> - We have refined Figure 2 by grouping metrics under "geometric / tuning / linear accessibility" and explicitly labelling each metric's focus on specific representational aspects within the figure.
>
> We appreciate the reviewer’s request for clearer motivation. We fully agree that the need for metric integration must be explicitly justified. In the revision, we now motivate this in the Introduction. Prior work has repeatedly shown that different representational similarity metrics yield complementary judgments about model relationships. For example, Bo et al. (2024) demonstrate that tuning-sensitive metrics such as soft-matching are far better at distinguishing architectural families, whereas geometry-preserving metrics (e.g., CKA/RSA) excel at separating trained from untrained models but conflate differences across architectures. Similar discrepancies between metrics appear in Kornblith et al. (2019) and Cloos et al. (2024). These results indicate that no single metric provides a complete or robust characterization.
>
> Without effective fusion, researchers must pick which invariances to privilege, which can lead to metric-specific conclusions or cherry-picked results. SNF addresses this by integrating complementary signals across metrics, yielding a consensus structure that is more stable, less sensitive to any individual metric’s invariance properties and more reliable for downstream analysis.
> We have acknowledged this explicit motivation for metric fusion in the revised Introduction.
>
> Refs:
>
> Bo, Y., Soni, A., Srivastava, S., & Khosla, M. (2024). Evaluating representational similarity measures from the lens of functional correspondence. arXiv preprint arXiv:2411.14633.
>
> Cloos, N., Li, M., Siegel, M., Brincat, S. L., Miller, E. K., Yang, G. R., & Cueva, C. J. (2024). Differentiable optimization of similarity scores between models and brains. arXiv preprint arXiv:2407.07059.
>
> Kornblith, S., Norouzi, M., Lee, H., & Hinton, G. (2019). Similarity of neural network representations revisited. In Proceedings of the 36th International Conference on Machine Learning (ICML) (pp. 3519–3529).

---

> ### Author Response · Authors · 2025-11-22
>
> ## R4-Q1: Circular reasoning?
>
> As illustrated above, the family label is used solely as a predefined metric for validation, rather than for regulating or selecting SNFs. The SNF approach itself does not depend on any family label.
>
> ## R4-Q2: Functional validation?
> As previously discussed in response to R4-W-Major 2, this is beyond the scope of this paper.
>
> ## R4-Q3: Why use average pooling rather than the CLS token?
>
> Our choice of average pooling was made for consistency across architectures: because CNNs naturally produce spatial feature maps that are typically summarized via global average pooling, we applied the same operation to ViTs to ensure that all models were compared under a unified embedding procedure rather than mixing architecture-specific conventions.
> This choice is also supported empirically. Dosovitskiy et al. (2021, App. D.3) report that a classifier trained on average-pooled ViT patch embeddings performs comparably to the standard CLS-token classifier, indicating that simple averaging preserves task-relevant information and yields representations similar in quality to the CLS token.
>
> ## R4-Q4: How would the classification change if web-scale models were included?
>
> We anticipate that the introduction of web-scale and multimodal encoders will expand rather than overturn existing typologies.
>
> ## R4-Q5: Differences from existing clustering models; is SNF the sole innovation?
>
> We thank the reviewer for raising this question. To our knowledge, none of the cited works - Kornblith et al. (2019), Ding et al. (2021), or the Rosenfeld reference (we would appreciate clarification on which specific Rosenfeld paper is intended) - propose a framework for constructing a model typology or aggregating multiple representational similarity metrics. For example, Kornblith et al. (2019) evaluate individual similarity metrics (e.g., CKA) but do not compare a broad range of metrics, and do not aim to build a taxonomy of models.
>
> Our core contributions are as follows:
>
> - We evaluate a diverse set of representational similarity metrics across a wide range of modern vision models, quantifying how well each metric distinguishes known model families. In the revised manuscript, we also evaluate each metric’s ability to recover meaningful structure in biological neural data (e.g., matching homologous visual areas across subjects, and reproducing the ventral-stream hierarchy). To our knowledge, such a comprehensive and granular multi-metric comparison spanning a diverse range of metrics has not appeared in prior work.
>
> - We treat each metric as providing a distinct “perspective” on representational structure and use Similarity Network Fusion to integrate these complementary signals into a consensus similarity matrix. SNF is a key methodological contribution: it provides a principled way to combine metrics with different invariance properties, reducing sensitivity to any single metric and yielding more stable and interpretable clusters.
>
> - Because our framework evaluates both individual metrics and their fusion, we are able to identify consistent representational trends (e.g., SSL models clustering across architectures under matched training data).
>
> Taken together, SNF is not the only novelty. The paper also contributes a broad multi-metric evaluation framework, a set of empirical findings about metric behavior and model families, and new validation analyses using biological neural data.
>
> ## R4-Q6: Why does the similarity within the VGG family appear weaker than that within the ResNet family in most metrics?
>
> We concur that this is an intriguing phenomenon, though it is not the focus of this paper. A plausible explanation is that the ResNet family shares residual connections, enabling deeper ResNets to achieve nearly identity mappings across a broad range of layers. Consequently, ResNets of differing depths remain closer in the representation space. In contrast, VGG lacks residual pathways, and increasing depth often leads to significant changes in the geometry of representations.

---

> > ### Comment · Reviewer_tjUP · 2025-11-27
> >
> > I thank the authors for their detailed responses and for the additional clarifications. The revision addresses several concerns, and I appreciate the effort put into the rebuttal. Below I summarize how my concerns were resolved and note a few points where I remain partially unconvinced.
> >
> > **1. Circular reasoning / interpretive coupling**
> >
> > I appreciate the clarification that SNF is not tuned using family labels, and the biological validation strengthens the case for SNF's validity.
> >
> > My original concern was more about interpretive coupling, rather than algorithmic leakage: metrics are evaluated on pre-defined families, SNF is selected for high performance on this criterion, then SNF is then used to derive a "data-driven typology," which unsurprisingly reproduces those most of the families.
> >
> > The rebuttal makes clear that the authors view family labels as a pragmatic external reference, which I accept. However, some of the manuscript's phrasing (e.g., "reveals the true 'species' of vision models") still feels strong given that the typology is unavoidably shaped by the metric suite used during fusion. The revised discussion appropriately notes this limitation, and I encourage maintaining this more tempered framing throughout.
> >
> > Overall, I am mostly satisfied, though I still regard the interpretability of the taxonomy as constrained by this evaluation setup.
> >
> > **2. Linear-accessible-information metrics**
> >
> > I appreciate that the authors acknowledge the limitations of correlation-based metrics and note this as an "open question."
> >
> > However, the manuscript still interprets weak discrimination as potentially suggesting that "this aspect of representation is more consistent across model families", while declining validation as "out of scope". My concern is that making interpretive claims while simultaneously acknowledging they may be invalid and declining to validate them does not meet expected rigor standards. This limitation should be discussed more substantially. The current treatment suggests more confidence in the interpretation than the evidence supports (abstract).
> >
> > **3. Model scope**
> >
> > I find the authors’ justification for restricting to ImageNet-1k models scientifically valid and appropriate for isolating architectural vs. objective effects. I also appreciate the added discussion explicitly stating this rationale and acknowledging that extending the analysis to web-scale/multimodal encoders is an important future direction.
> >
> > My only remaining reservation is practical: for many researchers, the models included are no longer central to current practice, so the resulting typology may be less broadly actionable.
> >
> >  **4. Presentation and terminology**
> >
> > The revisions generally improve clarity, and I appreciate the reorganization of figures.
> >
> > Some inconsistencies remain:—for example, figures still label several models as “unsup.” despite the claim of harmonized terminology.
> >
> > - On terminology, while the text now uses “self-supervised” consistently, the figures (2, 4, 5, and several appendix plots) still include parenthetical “unsup.” labels. This is a minor inconsistency but undermines the authors’ claim that the manuscript was fully harmonized.
> > - The revised Figure 2 is clearer with the metric-class grouping, though ordering metrics of the same class adjacently would further increase interpretability.
> >
> > This is minor but worth correcting.
> >
> > **5. Novelty and contribution**
> >
> > The authors' clarification of contributions, i.e., comprehensive multi-metric evaluation, SNF fusion, and cross-dataset/neural-data validations, is helpful and the framing is now clearer.
> > However, the practical implications remain under-developed. What would researchers actually do with this typology? Does typological similarity predict transfer learning performance? Does diversity improve ensembles? Could it guide model selection or architecture design?
> > A brief discussion of concrete applications would substantially strengthen the work. Without demonstrated practical utility, i.e., especially given the ImageNet-1k scope limitation, the contribution's impact remains uncertain.
> >
> >
> >
> > Given the improvements, the added analyses, and the clarifications, I increase my score from **2 → 4**. The work is technically sound with valuable elements (particularly the biological validation), but falls short of the thoroughness and practical impact expected at this venue. I could be persuaded toward (borderline) acceptance if the practical value, and unvalidated claims issues are addressed.

---

> > > ### Author Response · Authors · 2025-12-04
> > >
> > > We greatly appreciate your response, but we would still like to seek further clarification on certain issues.
> > >
> > > ## Circular Reasoning / interpretive coupling
> > >
> > > We can omit the word “true” to soften our expression.
> > >
> > > ## Linear-accessible-information metrics.
> > >
> > > Our research focus is not on linear-accessible-information metrics. The main contribution has a weak correlation with that, as we are evaluating more than just linear-accessible metrics. We are also not proposing a new linear-accessible similarity metric. Furthermore, the statement that “this aspect of representation is more consistent across model families” is based on our observed results, and we  have used “might suggest” to soften our statement.
> > >
> > > ## Model scope
> > >
> > > Thank you very much for your feedback regarding model scope. We will incorporate more up-to-date models in the future.
> > >
> > > ## Presentation and terminology
> > >
> > > We have modified the label in the figures to “selfsup.”
> > >
> > > Figure 2 is ordered as SNF, baseline, and single metrics sorted (from highest to lowest) by the average of the silhouette score/contrastive ratio in the heatmap, thereby providing readers with the corresponding metric performance details in Figure 3.
> > >
> > > ## Novelty and contribution
> > >
> > > Our research focuses on representation similarity, while the “practical implications” are actually closer to behavior/functional similarity, which is another type of model similarity metric. The relationship between behavior similarity and representational similarity requires further investigation and has been proposed as a complementary desideratum for evaluating representational similarity metrics (Bo et al., 2025; Ding et al., 2021). We will point this out in the conclusion.

---

### Official Review · Reviewer_XMWv · 2025-11-01

**Soundness:** 2
**Presentation:** 2
**Contribution:** 2
**Rating:** 4
**Confidence:** 3

**Summary:**

The paper takes inspiration from the field of biology to build a taxonomy of vision models based on their similarity with respect to the various metrics, similar to how taxonomy of biological organisms are build based on various genetic, phenotypical and morphological similarities. The authors posit that such a taxonomy of model can reveal similarities in learned representation that can transcend architectural and training objective differences. Thirty-five vision models were categorized into six categories based on training style and architecture, and their similarity computed with several metrics. An approach of combining the metrics that directly mirrors approaches used in biology has been presented.

**Strengths:**

1.	Biology inspired approach for hierarchically clustering vision models similar to building a taxonomy of biological species.
2.	Hierarchical clustering to produces a typology that transcends architecture labels. For example, the finding that self-supervised CNNs cluster with self-supervised ViTs rather than with their supervised CNN counterparts shows that training objective can take precedence over the architecture in determining the learned representation.
3.	Extensive experiment covering wide variety of vision models and similarity metrics.

**Weaknesses:**

1.	No discussion on how to analyze SNF based taxonomy. How can a researcher know which aspect of two models are contributing to similarity without looking at individual metrics and just from SNF? If I need to look at individual metrics, than what is the need of SNF? For a researcher who analyzes the reason for the spurious similarity between MAE trained ViT and fully supervised ConvNext, what kind of lead does SNF provide, beyond just telling that there is some similarity?
2.	SNF’s similarity score is not interpretable to understand how each individual metric is contributing to the overall score. This also relates to the previous point of how a researcher can make use of the score for further analysis.

**Questions:**

Please refer to the weaknesses section

---

> ### Author Response · Authors · 2025-11-22
>
> ## R3-W1: How does one conduct analysis based on the SNF taxonomy? What additional information does SNF provide?
>
> We appreciate the reviewer’s question about interpretability and how researchers should use an SNF-based taxonomy in practice. We agree that SNF is not designed to replace individual metrics or to obscure their contributions. Instead, SNF serves a different and complementary purpose: it provides a stable, consensus estimate of representational organization when individual metrics disagree, while still allowing researchers to inspect each metric separately when they wish to understand why two models are similar. Different metrics emphasize different invariances (geometric, subspace, tuning-curve-level correspondence, linear predictability, etc.) and often yield conflicting judgments. SNF does not claim to make these invariances disappear; rather, it integrates the shared signal across metrics to give a robust, global structure of model relationships, one that is less brittle to metric choice and far more stable across datasets and families. This consensus structure is valuable for tasks such as identifying broad model families, detecting stable clusters, and guiding sampling for downstream experiments (e.g., probing which models are “representationally near” one another).
>
> If a researcher wants to understand why two models (e.g., MAE–ViT and supervised ConvNeXt) appear close under SNF, the workflow we advocate is: (i) use SNF to establish that the similarity is consistent across metrics (i.e., not an artifact of a single measure), and (ii) then examine the individual similarity components (e.g., soft-matching vs. RSA vs. CKA) to understand which aspects of the representations align. SNF therefore acts as a starting point, not an endpoint. It ensures that follow-up investigation is grounded in similarities that are genuinely convergent across different representational axes, rather than artifacts of a single metric’s invariance properties. We now clarify this intended usage in the Discussion.
>
> SNF is not meant to produce a single interpretable scalar per se. Its purpose is to yield a more reliable topology of the representational space, i.e. a structure that captures model–model relationships robustly. This is exactly analogous to ensemble methods in machine learning: the ensemble prediction is stable and useful even if each component captures different signals. Researchers still have access to the components when they want fine-grained interpretability.
> We also highlight that our work contributes beyond SNF-fusion alone. We also systematically evaluate each metric’s ability to recover meaningful distinctions among model families and now among biological visual areas using NSD (new in the revision; see appendix I). This provides the first large-scale empirical comparison of commonly used metrics against an independent biological hierarchy, offering clear guidance for future representational studies.
>
> ## R3-W2: SNF’s similarity score is not interpretable to understand how each individual metric is contributing to the overall score.
>
> This is a natural property of SNF based on the graph diffusion mechanism: the final similarity is not a simple linear weighted sum but is obtained through multiple rounds of iterative updates. We view this as a trade-off between "reduced interpretable linear decomposability" and "enhanced robustness." In practice, we recommend the following approach: use SNF to identify "strong relationships" consistent across multiple metrics (such as long-range nearest neighbors and stable clusters), then return to individual metrics to examine their respective interpretations.

---

### Official Review · Reviewer_aKaZ · 2025-11-01

**Soundness:** 3
**Presentation:** 2
**Contribution:** 2
**Rating:** 2
**Confidence:** 4

**Summary:**

The authors propose the use of existing representation similarity metrics to compare the representations generated by various vision models. Furthermore, they propose Similarity Network Fusion (SNF), which combines existing metrics in a way that improves consistency and discriminative power.

**Strengths:**

•	The main claim of the paper, that similarity metrics (and specifically SNF) display interesting clustering patterns of architectures and training paradigms, is supported well. The experiments demonstrate higher separation between families with SNF compared to other metrics, and qualitatively SNF seems to be the best metric.
•	The authors clearly justify the gap that the proposed method is filling. Existing methods are inconsistent or fail to consider multiple types of representation structure, and the proposed method addresses these problems directly.
•	The writing is largely very clear and direct, with little confusion in most of the paper. /
•	With proper justification of its importance, the proposed method may provide a notable improvement over existing methods.

**Weaknesses:**

•	While the proposed work seems to glean interesting insights into model representations, it is not clear what the broader impact of this work is. Little time is spent justifying the importance of the method or possible applications. Most taxonomies in other fields (biology, linguistics, etc.) are an organizing means to an end to understand something about the underlying domain that the organization structure reveals, such as whether language similarity follows geographic proximity or evolutionary mechanisms and functional traits in biology. In the case of the proposed work, it is not clear what the taxonomy reveals (or would reveal) about the models’ assumptions, limitations, and applicability. Either theoretically or experimentally, showing the taxonomy improves model selection and performance for a task would better demonstrate its impact.
•	Unclear communication in critical parts of the paper, such as the methods section or when discussing the merits of the proposed method, hinder the possible impact of the paper.
      o	While the work overall is clear and straightforward, the presentation is hurt by some specific lack of clarity and sloppiness, particularly in figures and methods.
      o	Figure 3 has inconsistent y-axis scales, obscuring the fact that the separability metrics have very different values between datasets.
      o	Other figures are referenced with somewhat misleading descriptions, such as Figure F1. This figure is used to argue that the proposed method has some of the best performance on the CCC metric, but this is a stronger claim than the figure implies. Several metrics have similar performance, and the proposed method is not the top performing method in any of the given categories.
      o	In the methods section, the equations detailing the implementation of SNF are not described in detail, making it more difficult to understand the proposed methodology. Additionally, critical parts of experimental design are omitted from the main manuscript, such as how representations are generated for varying types of models.

**Questions:**

-	Figure F.1 seems to imply that several methods (specifically RSA and SoftMatch) have high cophenetic correlation coefficients across all four datasets. What does this imply about the proposed method? What does this imply about existing methods? Is this metric a good measure of the usefulness of a metric?
-	When discussing separation ability metrics, there is a specific focus on model “families”. What defines a family in this case: training paradigm, architecture, both, or neither?
-	In the paper, there is a brief mention of possible uses for such metrics. Are there cases when similar metrics have been used to analyze real-world behavior in contrast with metric similarity? Are there cases where similar metrics have been applied in other ways?

---

> ### Author Response · Authors · 2025-11-22
>
> ## R2-W1: What exactly does taxonomy reveal?
> We thank the reviewer for pushing us to clarify the broader impact and downstream value of a representational taxonomy. We agree that a taxonomy should ultimately serve as an “organizing means to an end,” not an end in itself, and we have revised the discussion to make these use cases more explicit.
>
> First, our framework provides a principled notion of a reference class for vision models. Much current work makes general claims like “self-supervised models learn X” or “ViTs behave differently from CNNs” based on a handful of hand-picked models, with no clear rationale for which other models those claims are meant to generalize to. By building a data-driven similarity space over many architectures and objectives, our SNF-based taxonomy identifies coherent “neighborhoods” of models that share representational structure. This is useful both scientifically and practically: it tells you when a mechanistic or behavioral finding is likely to generalize (because it holds across a cluster) and when it is likely idiosyncratic (because it occurs in an isolated or sparse region of the space). In the revision, we now emphasize this “reference class” role more clearly in the Discussion.
>
> Second, the taxonomy directly informs experimental design and model selection. When one wants to test whether a mechanistic claim, a robustness effect, or a brain-alignment result is general, our typology provides a principled way to: (i) sample diverse models across clusters to probe the robustness of a claim across representational “types”, (ii) sample redundant models within a cluster to estimate variance within a representational type, analogous to using multiple random seeds, and (iii) avoid over-interpreting results that only hold in a small, atypical region of the space.
>
> For practitioners, clusters in this space are natural candidates for predicting similar transfer behavior: models that are close in SNF space are expected to share inductive biases and thus to behave similarly under fine-tuning. While a full benchmark linking SNF distance to downstream task performance is beyond the scope of this paper, we now explicitly articulate this as a key application and natural next step.
>
> Third, the taxonomy reveals which design choices actually shape representations, and therefore constrains hypotheses about model assumptions and limitations. For example, we show that self-supervised models often cluster across architectural families, indicating that the learning objective can dominate architecture in determining representational structure, whereas in other regimes architecture remains the primary organizing axis. This is precisely the kind of structure that, in other domains, taxonomies are used to uncover: it tells us which constitutive properties (architecture, supervision, regularization) are actually predictive of emergent representations and where our coarse verbal labels (“ResNet”, “ViT”, “self-supervised”) are too crude.
>
> Finally, we agree that a compelling taxonomy should connect to an independent target beyond the models themselves. For this reason, we added a new analysis using brain data (Natural Scenes Dataset) to show that SNF is not merely organizing an arbitrary space: its structure is more predictive of known biological organization. Specifically, compared to single measures, SNF yields (i) far superior separation of early vs. higher visual areas and (ii) substantially higher recovery of the known ventral-stream hierarchy according to an anatomically motivated hierarchy score (see appendix I). In other words, the same representational organization that groups artificial models into sensible families also best preserves canonical structure in the biological visual cortex. We see this as an important proof-of-concept that our taxonomy is not just aesthetically pleasing but tracks a meaningful external criterion.
>
> ## R2-W2: The methodology section and key arguments are not sufficiently clear.
> We would appreciate it if you could be more specific about the points that need to be clarified.
>
>
> ## R2-W3: Graphical issues (inconsistent vertical axis scale in Figure 3; overly assertive wording in Figure F.1)
> We appreciate the suggestions and have made the following changes:.
> For Figure 3, we would plot a separate barplot for each dataset, with three subplots per plot, and provide the vertical axis scales for each subplot.
> For Figure F.1, the purpose of this figure is to demonstrate and validate the good quality of the clustering diagrams themselves.

---

> ### Author Response · Authors · 2025-11-22
>
> ## R2-Q1: RSA and SoftMatch also achieve high CCC across all datasets. What does this signify?
> As demonstrated in the main text, the CCC is mainly to show that the clusters formed by the fusion matrix more faithfully reflect the underlying similarity structure among models, showing the justifiability of the clustering operation. CCC panel should not be interpreted as a definitive evaluation of metric quality. In fact, CCC is not a meaningful or well-aligned criterion for assessing representational similarity metrics in our context—it does not tell us whether a metric actually captures meaningful distinctions between model families. The key criteria are the three complementary measures we report—d′, silhouette score, and contrastive ratio—each of which directly quantifies how well a metric distinguishes known model families. These metrics probe different aspects of discriminability (separation, cluster tightness, and signal-to-noise in family-level structure). They are therefore much more appropriate indicators of whether a similarity measure captures meaningful representational structure. Across all analyses, SNF achieves strong performance simultaneously on all three discriminability criteria, which is the central motivation for metric aggregation. We also note that the relative ranking of individual metrics is largely stable across these criteria, further reinforcing the robustness of our evaluation.
>
> ## R2-Q2: What is the definition of "family"?
>
> We define each family by the interaction between the training paradigm and base architecture (detailed in **Methods** and **Appendix B**). The main text has been updated to state this more explicitly.
>
> ## R2-Q3: Have such metrics been employed in analyzing real-world behavior?
>
> We appreciate the reviewer’s question about how these similarity metrics have been used in real scientific or applied settings. Indeed, representational similarity metrics are widely used across neuroscience, cognitive science, and machine learning, often to link model representations to real-world behavior or biological data. Similarity metrics are routinely applied to understand layer-to-layer transformations within a model or to compare different random seeds or training regimes. A large literature uses RSA-style metrics to compare deep neural networks with neural data from fMRI, ECoG, MEG, and single-unit recordings (e.g., Yamins & DiCarlo, Kriegeskorte et al., Cichy et al., Schrimpf et al.). Across all of these domains, researchers typically choose one or two metrics (often RSA or CKA), even though different metrics emphasize different aspects of representation structure. Our contribution is not to introduce a new use case but to show how multiple complementary metrics can be aggregated into a more stable and comprehensive signal and to benchmark their ability to recover meaningful distinctions in both model families and biological recordings. This is the gap we aim to address.
>
> Refs:
>
> Cichy, R. M., Khosla, A., Pantazis, D., Torralba, A., & Oliva, A. (2016). Comparison of deep neural networks to spatio-temporal cortical dynamics of human visual object recognition using MEG and fMRI. Scientific Reports, 6, 27755.
>
> Kriegeskorte, N., Mur, M., & Bandettini, P. A. (2008). Representational similarity analysis – connecting the branches of systems neuroscience. Frontiers in Systems Neuroscience, 2, 4.
>
> Schrimpf, M., Kubilius, J., Hong, H., Majaj, N. J., Rajalingham, R., Issa, E. B., … DiCarlo, J. J. (2018). Brain-Score: Which artificial neural network for object recognition is most brain-like? bioRxiv, 407007.
>
> Yamins, D. L. K., Hong, H., Cadieu, C. F., Solomon, E. A., Seibert, D., & DiCarlo, J. J. (2014). Performance-optimized hierarchical models predict neural responses in higher visual cortex. PNAS, 111(23), 8619–8624.

---

### Official Review · Reviewer_2rRX · 2025-11-03

**Soundness:** 3
**Presentation:** 3
**Contribution:** 2
**Rating:** 4
**Confidence:** 3

**Summary:**

The authors look at different metrics to quantify representational similarity between computer vision models. In particular Singular Vector Canonical Correlation Analysis (SVCCA), Projection-Weighted Canonical Correlation Analysis (PWCCA), Linear CKA, RSA, Soft Matching, Procrustes Alignment and Linear Predictivity are all computed and then combined with Similarity Network Fusion, a method inspired by multi-omics integration. The study clusters a range of 35 different vision models spanning different architectures (CNN vs ViTs) and objectives (SSL vs. supervised). By using various separation/clustering metrics, the authors compare their SNF metric with simple averaging of the metrics and each metric individually. The SNF approach recovers best the grouping by families and clusters SSL models together putting more emphasis on the objective in this case rather than the architecture.

**Strengths:**

* The paper studies an important question and applying Similarity Network Fusion to combining multiple representational metrics is novel as far as I can tell.
* The approach is clearly described and the paper is well motivated.
* The study investigates a broad range of different representational similarity metrics.

**Weaknesses:**

* The overall diversity of models considered for the analysis is quite limited. The analysis misses image/text models which are a widely used category of vision models and would add a fundamental new objective to the analysis. Further, only SSL models which have been trained on ImageNet-1k have been considered. Given the finding that SSL models cluster together over different architectures in the paper, it would be particularly interesting to see whether this also holds over different pretraining datasets. There are multiple publicly available SSL models which have been trained on other datasets (e.g. [1, 2]) which could be used in this study.

* The paper only considers natural image datasets: ImageNet-1k, CIFAR10, CIFAR100, Ecoset for computing similarities. Previous work [3, 4] has shown that similarities between models depend on the dataset which is used to compute the representations. A thorough investigation of representational similarity should go beyond ImageNet and also consider e.g. fine-grained and more structured datasets.

* There is not really a clear ground truth structure of which models should be closer together than others. On the one hand the paper is defining some families and evaluating the clustering based on this definition. But on the other hand, it is also noted that SSL models cluster over architecture families, basically asking the question whether we can even benchmark the right organization of similarity. I think the “desired similarity” should be more clearly described in this paper and the problem of benchmarking it further discussed.

[1] Oquab, Maxime, et al. "Dinov2: Learning robust visual features without supervision." TMLR 2024.

[2] Goyal, Priya, et al. "Self-supervised pretraining of visual features in the wild." arXiv preprint arXiv:2103.01988 (2021).

[3] Cui, Tianyu, et al. "Deconfounded representation similarity for comparison of neural networks." Advances in Neural Information Processing Systems 35 (2022): 19138-19151.

[4] Ciernik, Laure, et al. "Objective drives the consistency of representational similarity across datasets." ICML 2025.

**Questions:**

* What do the colors in the dendrogram in Figure 5 represent?

* Were the family definitions in Appendix B used to compute the separation ability metrics or how were these exactly computed?

---

> ### Author Response · Authors · 2025-11-22
>
> ## R1-W1: Limited model diversity, lacking non-ImageNet SSL and text-image models
>
> We concur with the reviewer's suggestion: extending the analysis to web-scale and text-image models represents a highly valuable direction. In this paper, we deliberately restricted the model pool to encoders pre-trained on ImageNet-1k. The primary aim was to compare the impact of different model families (defined by training objectives and architectures) on representations within a unified data and label space, thereby avoiding training data distribution itself becoming major confounding factors. Recent work (e.g., Conwell et al., 2024) shows that “visual diet” can be the main driver of representational differences, often overshadowing architectural or objective-level factors, and our design follows that observation. It is precisely this controlled setting that allowed us to observe that, under identical data conditions, SSL models cluster together across architectures.
>
> Incorporating web-scale SSL or image-text models such as DINOv2/CLIP directly would simultaneously alter both training objectives and data distribution, rendering the clustering structure harder to interpret: what exactly drives the differences—the data, the objectives, or the architecture? In the revised version, we have articulated the rationale behind this experimental setup more clearly (in the methods section; highlighted in red) and designated extending it to web-scale/multimodal models as a significant direction for future work (see the Discussion).
>
>
> ## R1-W2: Utilizes only natural image datasets, lacking finer-grained and structured datasets
>
> In this work, our objective was to conduct a systematic comparison of representation similarity metrics within the most prevalent scenarios in computer vision. For this reason, we use natural-image datasets such as ImageNet-1k, CIFAR-10/100, and Ecoset which already differ substantially in resolution, category granularity, and distribution. Existing work has indeed demonstrated that similarity depends on the dataset used to compute representations, which precisely motivates our analysis across multiple datasets rather than focusing solely on ImageNet.
>
> It is also worth noting that large-scale studies of representational metrics (e.g., Kornblith et al., 2019; Morcos et al., 2018; Raghu et al., 2017) typically evaluate on a single natural-image dataset, most commonly ImageNet or sometimes even just CIFAR-10. Our analysis therefore goes beyond the standard practice by assessing stability across four distinct natural vision datasets. Across all of them, both the metric rankings and the SNF-derived typology remain highly consistent, indicating that our main conclusions are not tied to any single dataset.
>
> We fully agree that probing how representational typologies shift under more structured, fine-grained, or distribution-shifted datasets is an important next step, and we now note this explicitly as future work in the discussion.
>
> Refs:
>
> Conwell, C., Prince, J. S., Kay, K. N., Alvarez, G. A., & Konkle, T. (2024). A large-scale examination of inductive biases shaping high-level visual representation in brains and machines. Nature Communications, 15, Article 9383. Published 30 October 2024. Nature
>
>
> Kornblith, S., Norouzi, M., Lee, H., & Hinton, G. (2019). Similarity of Neural Network Representations Revisited. In Proceedings of the 36th International Conference on Machine Learning (ICML), PMLR 97. Proceedings of Machine Learning Research
>
>
> Raghu, M., Gilmer, J., Yosinski, J., & Sohl-Dickstein, J. (2017). SVCCA: Singular Vector Canonical Correlation Analysis for Deep Learning Dynamics and Interpretability. arXiv:1706.05806. arXiv
>
> Morcos, A. S., Raghu, M., & Bengio, S. (2018). Insights on representational similarity in neural networks with canonical correlation. arXiv:1806.05759.

---

> ### Author Response · Authors · 2025-11-22
>
> ## R1-W3: Ideal similarity structure/ground truth structure remains unclear
>
> We agree with the reviewer that for artificial neural networks, there exists no singular "ground truth" similarity structure. Indeed, a central theme of our work is that the absence of a gold standard makes it fundamentally difficult to evaluate representational similarity metrics themselves. In our current work, we employ the "architecture + training paradigm" family label as a pragmatic proxy for an expected structure: if a metric cannot reliably distinguish even obvious differences—such as between ResNet and ViT, or supervised learning and contrastive self-supervision—it is likely to be of limited value in constructing an informative model taxonomy. Importantly, these labels are not used by SNF, which remains fully unsupervised; they serve only as a sanity check that the resulting clusters are not arbitrary.
>
> To address the deeper issue the reviewer raises, namely, the lack of an independent benchmark, we additionally introduce validation based on biological neural data (see appendix I), where a well-established hierarchical organization is known from decades of studies. Using responses from 10 visual regions in the Natural Scenes Dataset (NSD), we compute region-by-region similarity matrices and evaluate how well each metric recovers the expected anatomical hierarchy. Across all separability criteria, SNF substantially outperforms individual metrics (Fig. I2A): it achieves a mean d′ of 21.45 (nearly 5× the next best metric) and yields the clearest and most coherent geometry when visualized via PCA (Fig. I.1). Early visual areas (V1→V4) form a smooth progression, and higher-level ventral regions appear in their expected positions—patterns that are markedly less well defined under RSA, CKA, or SoftMatch.
>
> To quantify this more formally, we compute a ventral-stream hierarchy score based on Kendall’s τ, comparing similarity structure against a known discrete hierarchy (V1→V2 →V3→ V4→high-level ventral visual stream). SNF achieves the highest hierarchy score (Fig. I2B), indicating that it best preserves both within-area coherence and the expected ordering between areas. These results provide an external point of reference, independent of any architectural or supervision labels, demonstrating that the structure recovered by SNF aligns more closely with established biological organization than the baselines. This strengthens confidence in SNF as a principled way to aggregate metrics in the absence of a perfect benchmark.
>
> ## R1-Q1: What do the colors in the dendrogram of Figure 5 represent?
> The colors in Figure 5 denote the cluster (formed by SNF) each model belongs to.
>
> ## R1-Q2: Is the family definition in Appendix B used for the separability metrics?
> Yes. The family definition in Appendix B is precisely the label we employ when calculating separability metrics.

---

### Meta-Review · Area_Chair_Ezez · 2025-12-21

**Summary:**

* The diversity of adopted model architectures and datasets is limited.
* There is no ground truth for such a taxonomy construction of different architectures.
* While a taxonomy of architectures is constructed, it is not explicitly demonstrated how this taxonomy can be further leveraged in downstream uses.
* The derived fused similarity is hard to interpret.

IMHO, the above major concerns are correlated to each other. When the investigated models are limited to those trained on ImageNet, it becomes unclear what the similarity between models represents and how it can be further leveraged. The resulting taxonomy seems to group similar architectures and training schemes, which is natural by instinct, but the underlying mechanism of such grouping is not well interpreted.

The positioning of this paper is actually similar to [1]. Comparatively, [1] provides empirical observation on a much larger scale. [1] also proposes some hypotheses for the convergence of model parameters and provides some insights for subsequent model developments. The authors are encouraged to discuss their differences from [1] and highlight their unique discoveries.

[1] Huh, Minyoung, et al. "Position: The platonic representation hypothesis." In ICML. 2024.

**Reviewer Concerns:**

The authors explained their motivation to conduct such an exploration. However, from my point of view, the concerns shared by reviewers have not been well addressed. The most significant concern is the downstream use or deeper analysis of the models' behaviors.

**Reviewer Scores:**

I do not think the concerns have been well addressed.

---

### Decision · Program_Chairs · 2026-01-26

Reject